# Machine learning discovers numerous new computational principles supporting elementary motion detection

Alon Poleg-Polsky ⬤ ✉

Motion direction detection is a fundamental visual computation that transforms spatial luminance patterns into directionally tuned outputs. Classical models of direction selectivity rely on temporal asymmetry, where motion detection arises through either delayed excitation or inhibition. Here, I used biologically inspired machine learning applied to retinal and cortical circuits to uncover receptive field architectures capable of direction selectivity. These include mechanisms based on asymmetric synaptic properties, spatial receptive field variations, new roles for pre- and postsynaptic inhibition, and previously unrecognized kinetic implementations. Conceptually, these circuit architectures cluster into eight computational primitives underlying motion detection, four of which are previously undescribed. Many of the solutions rival or outperform classical models in both robustness and precision, and several exhibit enhanced noise tolerance. All mechanisms are biologically plausible and correspond to known physiological and anatomical motifs, offering fresh insights into motion processing and illustrating how machine learning can uncover general principles of neural computation.

Visual sensation begins with photoreceptors, but they do not encode motion information. To perform motion computations, the visual system must transform spatially separated luminance signals to an output that is stronger in specific directions[1–4]. Elementary motion detectors, composed of shallow networks with directionally untuned inputs, represent the simplest neural substrate that can support the computation of direction selectivity (DS). Over 60 years ago, two foundational studies proposed temporally asymmetric circuitry as the basis for DS in elementary motion detection circuits. The core computation of the Hassenstein & Reichardt (H&R) detector is delay-and-compare, where signals from spatially offset inputs are multiplied after the input activated first in the preferred direction is delayed by a low-pass filter[5]. Similarly, Barlow and Levick's (B&L) model compares fast excitatory signals with delayed inhibition from the "null" side -- the position from which the null direction stimulus enters the cell's receptive field (RF) -- to suppress responses to motion in that direction[6]. While these classical models have been extended and refined by more recent research, revealing several distinct biological implementations, the core computational motifs of kinetic filtering applied to preferred side excitation or null side inhibition remain central to our understanding of motion detection[2,3,7–28].

The anatomical simplicity of elementary motion detectors belies their potential for remarkable functional diversity. The neurons in the input layer can process sensory stimuli in complex, nonlinear ways, depending on their filter characteristics and the statistical properties of the stimulus[29,30]. Accordingly, past research identified several distinct solutions to motion detection in biological circuits mediated by asymmetric filtering of sensory input through dissimilar synaptic dynamics and receptive field properties[8,9,22–24,31–37]. However, it remains unclear whether additional circuit-based solutions exist and to what extent known algorithmic implementations of directional selectivity encompass all potential operational principles of elementary motion detectors.

Department of Physiology and Biophysics, and Neuroscience Program, University of Colorado School of Medicine, Aurora, CO, USA.
✉ e-mail: alon.poleg-polsky@cuanschutz.edu

In this study, I employed machine learning to systematically explore how functional architectures of spatial and temporal filters of RF center and surround components are capable of supporting motion detection. Exploration of the vast combinatorial space formed by variability in individual RF components and their interactions revealed multiple distinct circuit configurations capable of performing motion computations with multiple asymmetries in the RF composition capable of leading to DS levels that were often comparable to those observed in the classical models. Notably, despite structural differences among these circuits, their information processing consistently converged on a small set of algorithmic implementations, which I refer to as 'computational primitives'. These included classical models like H&R and B&L, as well as previously unrecognized circuit motifs. Examination of these algorithmic implementations showed they can drive robust direction selectivity and may sustain enhanced noise tolerances. These findings highlight the power of machine learning to uncover previously unrecognized solutions and provide algorithmic-level insights into neural computations mediated by simple networks that often elude human intuition.

## Results

### Machine learning framework to study motion computations

The ability to detect motion direction was examined using feedforward, shallow network architectures. In the retina, previous studies have shown that excitatory inputs from bipolar cells to direction-selective ganglion cells (DSGCs) are organized into distinct populations arranged sequentially along the cell's preferred motion axis[38]. In this model, stimulus activation was first filtered through a linear spatial RF, where the overlap between the stimulus and the RF shape determined the potential for graded transmitter release. Temporal filtering, together with readily releasable pool (RRP) dynamics, then governed the actual rate of synaptic drive onto the postsynaptic neuron (see "Methods" for more details). To evaluate the motion detection capacity of this architecture, I conducted simulations in which four excitatory input populations -- each potentially having distinct receptive field properties -- were spatially distributed along an arbitrarily defined preferred direction axis onto a multicompartmental DSGC model (Fig. 1a). Only excitatory inputs were included in this initial set of simulations.

Consistent with biological observations that most bipolar cell terminals lack directional tuning[39], all presynaptic inputs in these models were set to produce similar responses to stimulation in different directions. Directional selectivity of the DSGC was quantified using a direction selectivity index (DSI), computed as the vector sum of peak subthreshold membrane potential at the soma across five stimulus velocities (Fig. 1a)[40,41]. While subthreshold responses generally show weaker directional tuning than spiking output[33,36,42–48], this approach allowed robust detection of DS mechanisms across a range of stimulus intensities without the confounding effects of spiking nonlinearities (Supplementary Fig. 1).

Machine learning was implemented as follows: starting with randomly initialized models, the fittest model (determined by the highest directional performance) seeded the next generation. The parameters describing the characteristics of presynaptic RFs were adjusted using a genetic algorithms-based machine learning paradigm[49,50]. In all cases, including the seed values, parameters were set to range within biologically relevant boundaries. In most simulations, constraints were put on the difference in RF description of the presynaptic inputs. Some parameters were allowed to vary independently across presynaptic groups. To explore the parameter space beyond the constraints of the initial seed values, other parameters were 'shared' or mutated jointly for all presynaptic cells. Overall, this approach enabled the identification of optimal configurations for directional tuning among presynaptic populations for various aspects of their RF structure.

To validate the effectiveness of this search strategy, I tested the ability of the algorithm to find the H&R solution for optimal spatio-temporal distribution of units with distinct kinetics. Evolved simulations ($n = 100$ independent runs) converged on a singular outcome where response dynamics became progressively faster with increasing distance from the preferred side of the DSGC, leading to synchronized peak responses and robust postsynaptic summation in the preferred direction (DSI = $23.8 \pm 1.6\%$, Fig. 1b$_i$), as postulated by the H&R model[5,26,38].

### Direction selectivity computation with differences in presynaptic receptive field sizes and orientations

The spatial RF properties can influence motion processing and impact signal transformation in H&R model families[31,32,49], but could they mediate DS computations in the absence of different temporal filters? To investigate whether and how inputs that only differ in spatial RF properties could compute DS, I evolved networks with presynaptic groups constrained to have identical kinetics but potentially variable extent of their RFs (Fig. 1c$_i$). In this configuration, all presynaptic neurons produced identical responses when stimulated by a full-field stimulus. However, the representation of moving bars was dissimilar, with larger RFs activated earlier, for longer duration and smaller amplitude, reflecting the slower engagement of the RF by the stimulus[49,51,52]. Evolved models demonstrated that differences in spatial RF properties can lead to strong directional tuning (DSI = $22.1 \pm 0.8\%$) and that the optimal functional circuit configuration that supported DS was formed with progressively increasing RF diameters in the input population along the motion preference axis (Fig. 1c$_i$, c$_{ii}$).

The spatial arrangement of the presynaptic RFs in the optimal solution offers an insight into how synchronized engagement of the presynaptic populations is achieved. RFs from the different input populations converge near the distal region of the postsynaptic dendritic arbor at the location where the preferred direction stimulus first encounters the cell (Fig. 1c$_{ii}$). This near-alignment facilitates motion computations that begin substantially earlier than in the kinetic (H&R-like) implementation (Fig. 1b–e, Supplementary Fig. 1)[53], potentially compensating for delays in retinal circuitry, a property that is important in circuits where speed of motion detection is paramount[54–56].

Spatial differences between presynaptic cells are not limited to size differences. Indeed, not all RFs are perfectly round, and orientation-selective neurons with elliptical RFs are frequently observed in DS circuits[36,57–61]. Implementing models with non-circular RFs revealed that DS computation can take advantage of presynaptic populations that vary only in their orientation. In the optimal configuration, the long RF axis of the presynaptic population transitioned from perpendicular to parallel along the preferred direction of motion. This arrangement of orientation-selective cells closely mirrored the RF size-based solution in terms of directional selectivity levels (DSI = $21.2 \pm 0.7\%$), velocity tuning, and detection time (Fig. 1d, e).

Both the kinetic and spatial solutions described above produced comparable selectivity to bars of varying widths and supported strong directional spiking responses when sodium and potassium channels were added to the postsynaptic soma (Supplementary Fig. 1). From an algorithmic perspective, organizations of the presynaptic spatial RFs promoted an 'anti-H&R' computation where neurons with slower motion responses, mediated by larger RF sizes or orientation of the long axis of an oval RF parallel to the preferred motion direction, innervated the null side of the postsynaptic cell and thus were recruited last in the preferred-direction activation (Fig. 1c–e).

To confirm that DS was driven by asymmetric presynaptic circuitry rather than postsynaptic voltage effects[62–70], I examined directional tuning in models where all input groups had identical properties. In these cases, the DSI of the best-performing models was only $2.4 \pm 0.1\%$, indicating a minimal contribution of postsynaptic processing to the computation (Fig. 1e, 'symmetric RF').

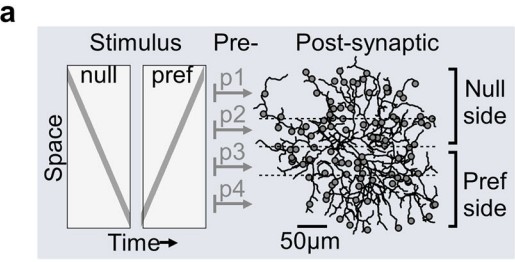

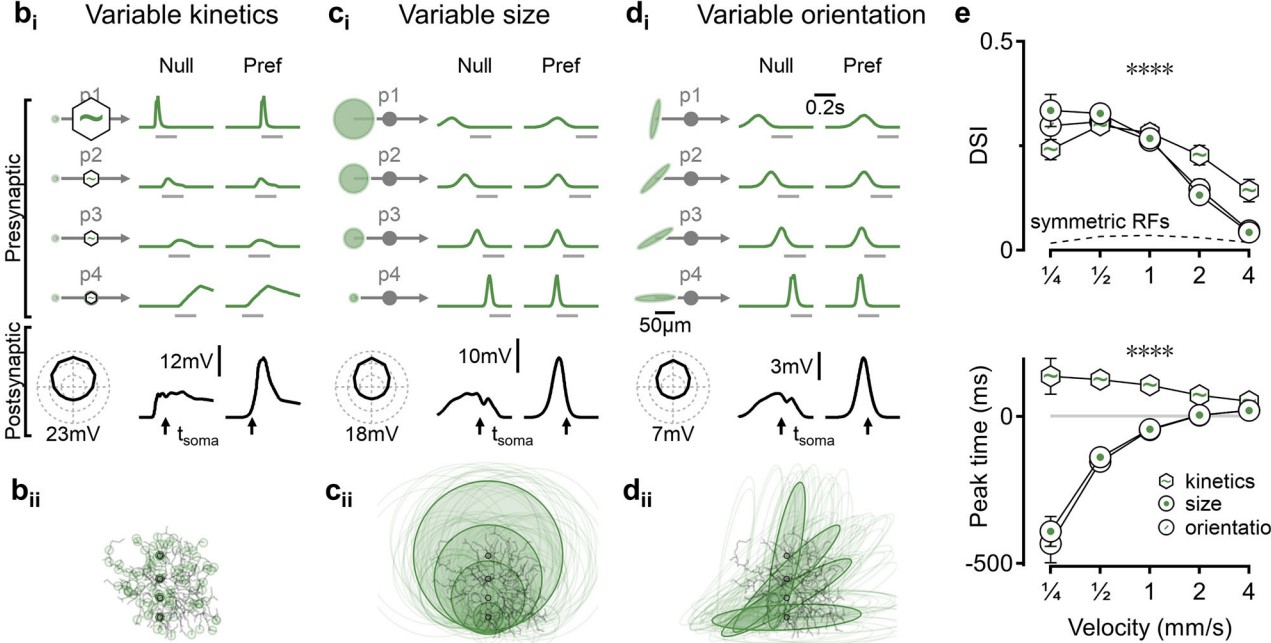

**Fig. 1 | Potent and early direction selectivity computation in circuits with varying spatial receptive fields. a** Schematic of the simulated circuit. Bars moving in different directions (left) activate neuronal populations with distinct receptive field properties, each synapsing onto a different region of the postsynaptic cell along the null-preferred axis (right, gray circles). The model is trained using a genetic algorithm-based approach to explore network architectures that promote directional tuning. **b$_i$** A representative solution for models with variable temporal dynamics in presynaptic populations. Top, normalized time courses of motion responses from a single cell in each of the four presynaptic groups (green), with horizontal gray bars indicating the stimulation period. Faster responses are marked schematically with larger hexagons. The optimal dynamics shifted from slow to fast along the preferred motion axis. Note that the presynaptic responses were direction-independent, and directional selectivity arises due to distinct temporal activation sequences. Bottom, right: postsynaptic somatic membrane potentials, t$_{soma}$

indicates the time point at which the stimulus reached the spatial position of the soma. Left, directional tuning calculated from 12 directions. **b$_{ii}$** Superimposed presynaptic receptive fields on the postsynaptic cell, for the model shown in b$_i$. Four representative RFs are shown in bold, one for each presynaptic population; their position is highlighted with a small circle. **c, d** As in (**b**), but with constant kinetics and varying RF sizes (**c**) or RF orientations (**d**) across presynaptic populations. e, Top, the mean (±standard deviation) velocity tuning of the model families presented in (**b–d**). The dotted curve represents a model with identical presynaptic receptive fields, where motion computations can only rely on postsynaptic processing. Bottom, time difference between the peak response in the preferred direction and the time the stimulus reached the postsynaptic soma (tsoma in **b–d**). $N = 100$ independently randomized simulation runs, ****$p < 0.0001$, two-sided $t$-test for each velocity between kinetic and spatial models with Bonferroni correction for multiple comparisons.

## Mechanisms of direction selectivity computation with presynaptic surrounds

The investigation of how presynaptic RF architecture influences DS computation would be incomplete without considering the effects of surround suppression -- a canonical property of most visually active cells[71] -- that is known to impact motion computations[14,72,73]. In the models presented below, the surround component was implemented using a separate spatiotemporal filter, spatially aligned with the center RF. Surround activation reduced the net synaptic drive: the full RF signal was computed as the center minus surround conductances and then rectified to prevent negative synaptic conductance. To systematically examine the role of presynaptic surrounds in DS, I simulated inputs with varying surround strength, kinetics, and spatial properties, including size and shape (Fig. 2).

Consistent with the larger number of parameters describing the center-surround RF, these surround models generated diverse

solutions, which fell into two main categories: circuits having preferred-side inputs with strong and fast surrounds that delayed and narrowed the full RF responses (Fig. 2a$_i$–c$_i$), or organization of presynaptic cells with stronger and slower surrounds on the null-side portion of the postsynaptic dendritic arbor (Fig. 2a$_{ii}$–c$_{ii}$). Functionally, although asymmetric surrounds could promote DS through H&R and anti-H&R computations, many solutions did not produce motion responses with sufficiently distinct kinetics, which is the cornerstone of these computations (e.g., Fig. 2c$_{ii}$). Instead, across many circuit configurations, surround suppression strategically blocked responses at specific time points to temporally align the peak waveforms of the full RF across all presynaptic inputs (Fig. 2b$_i$, c$_i$, and c$_{ii}$). Importantly, this temporal alignment occurred only during motion responses (Fig. 2d), as RF activation dynamics were shaped by the differential spatiotemporal engagement of center-surround components with the moving stimulus.

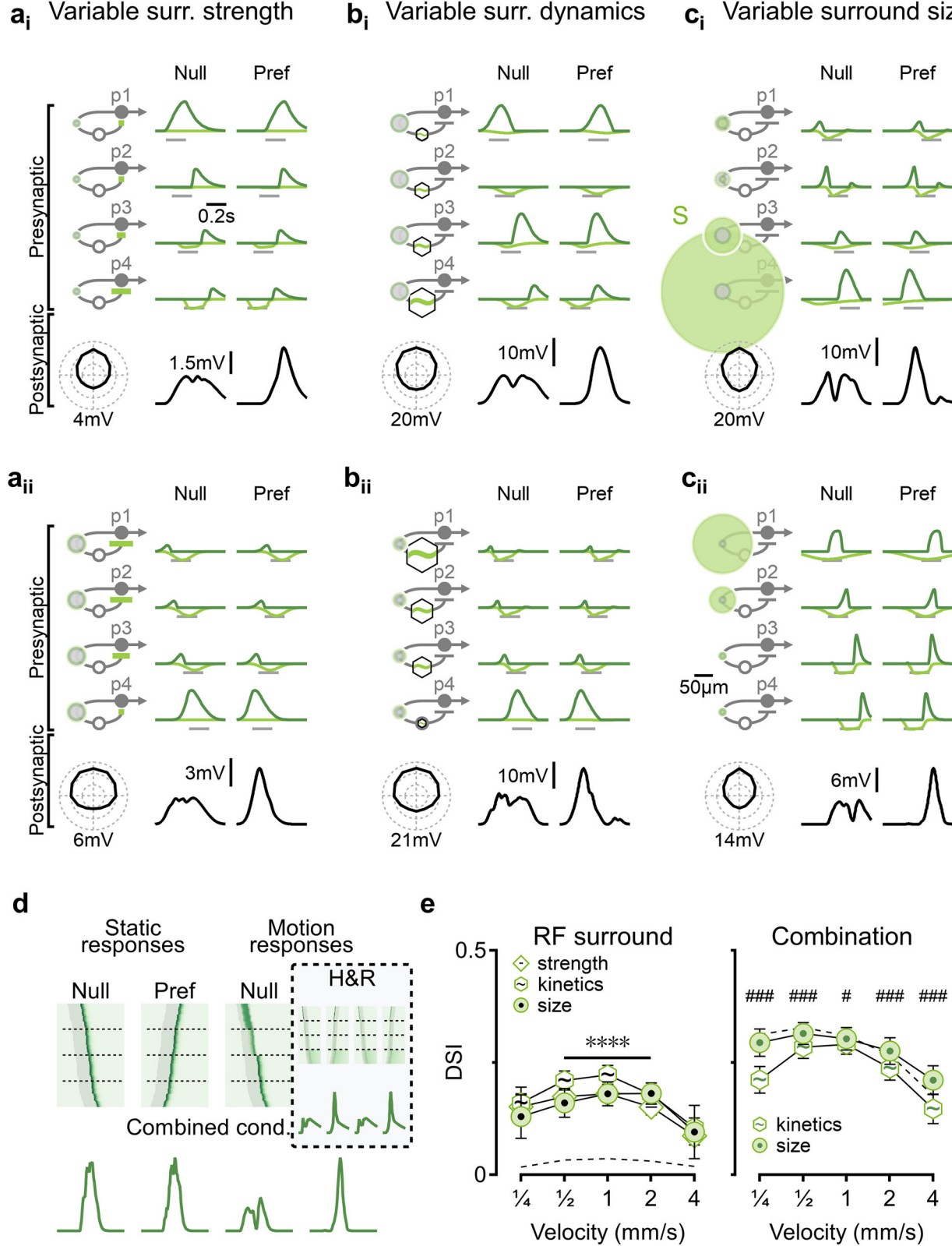

Overall, in models with variable surrounds, kinetic models achieved the highest DSI levels for moving bar stimulation (17.5 ± 1.4%, Fig. 2e) and for drifting gratings (Supplementary Fig. 2). However, combination of variable center and surround components produced stronger directional tuning (DSI = 28 ± 1.3%) in models with variable center and surround diameters compared to those with variable center and surround kinetics (DSI = 23.4 ± 0.9%, Fig. 2e), the latter being close to the performance of

models with free spatiotemporal centers (DSI = 24.6 ± 1%). These spatial solutions matched the directional performance of models with unconstrained center-surround properties (DSI = 27.9 ± 1%, Fig. 2e).

**Direction selectivity with synaptic weights**

Algorithmic implementations of DS computation rely on asymmetric processing across different information channels that feed into the

**Fig. 2 | Directional computation with presynaptic surrounds. a** Two distinct solutions (**i** and **ii**) were observed in models where presynaptic surround strength is the free parameter. Surround RF components are shown in light green. **b**, **c** Similar to (**a**), but with variation in dynamics (**b**) or spatial extent (**c**) of the presynaptic surrounds. **d** Presynaptic surrounds can enable DS by temporally aligning motion responses in the preferred direction. Left, activation profiles of synaptic inputs, sorted by vertical positions. Stimulation times are highlighted in gray. Bottom: cumulative postsynaptic conductance shows selectivity only for motion responses. Right: a similar analysis of a model with varying center dynamics (Fig. 1b) shows

comparable tuning for both static and moving stimuli. **e** Left: the mean (±standard deviation) velocity tuning in models from (**a–c**). Dashed, models with identical presynaptic center-surround RFs. Right: velocity tuning when both center and surround RFs vary. Dashed, an unconstrained model. $N = 100$ independently randomized simulation runs for each model. ****$p < 0.0001$, between different models. #$p < 0.05$, ###$p < 0.001$, between kinetic solutions and other models, using ANOVA for each velocity with Bonferroni correction for multiple comparisons. Error bars: standard deviation.

detector[1]. Prior work suggested that asymmetry in synaptic weights between the input population and the detector may be sufficient to promote directional capabilities[74]. To better understand the factors enabling a circuit with different synaptic weights to identify the sequence of presynaptic activation, I first examined this question in a minimal model that featured an input layer composed of two cells with identical filters that summed linearly at the detector (Fig. 3a). Already in this simple scenario, the summation of two inputs with similar kinetics but different amplitudes was found to be directionally asymmetric. The maximal signal was reached with sharply rising, slowly decaying responses when the weaker input was recruited first (Fig. 3a). In some models, an alternative solution with slower kinetics and reversed weights was reached, but the directional selectivity of this solution was about half of the optimal model (DSI = 6% vs. 11%, Fig. 3a, inset).

Encouraged by these results, I pursued the examination of weight-based solutions in the full DSGC model, probed at multiple velocities and directions (Fig. 3b, c). As its minimal counterpart, this biologically grounded circuit could be trained to generate directional preference, with the optimal solutions being a band-like organization of stronger synaptic inputs perpendicular to the motion preference axis (Fig. 3b). As in minimal models, two solution types were apparent. About half of the models evolved to have two bands of stronger synapses, with the band on the null side of the DSGC having more numerous inputs, mirroring the optimal solution seen in minimal models. In other cases, double strong-to-weak transitions evolved, each following the logic of the alternative minimal model solution (Fig. 3b, right). Directional tuning was observed in all simulations across multiple velocities; however, as expected, it was less robust compared to variable-RF models (DSI = 7.4 ± 0.4%, Fig. 3b, c). Importantly, however, this result indicates that differential spatiotemporal filters are not required for the detection of motion direction, demonstrating a potential for synaptic weight differences in implementing DS computations.

### Computational primitives in elementary motion detectors

To identify additional DS algorithms in elementary motion detection circuits, I modeled various configurations of excitatory cells where multiple RF and synaptic parameters were free to vary between the input groups (Supplementary Figs. 3, 4). Critically, all solutions converged on one of the four computational primitives revealed in this study, namely H&R, anti-H&R, amplitude, and temporal alignment (Fig. 4a). The probability of observing the computations in different functional circuits is illustrated as a two dimensional plot, in which the H&R and anti-H&R implementations, which rely on temporal differences in motion responses, form the horizontal axis. The vertical axis indicates the degree of similarity of normalized response waveforms in the preferred direction (temporal alignment) or at the time of stimulus onset (amplitude). Some solutions could be explained with a mix of primitives, for example, by forming H&R-like slower signals that are also delayed in time on the preferred side of the detector, as was often seen in circuits with varying surrounds (Fig. 4b).

This examination of the parameter space revealed that amplitude-based computational primitives yielded the lowest selectivity. This mechanism was optimal only in models where the free parameter was

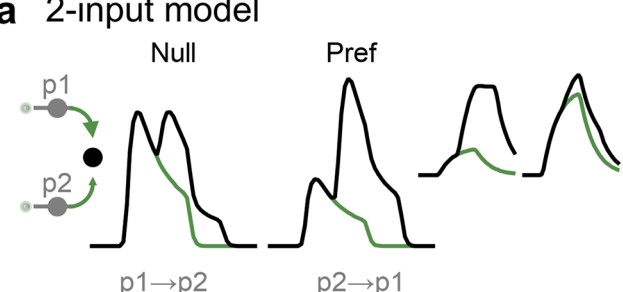

## a   2-input model

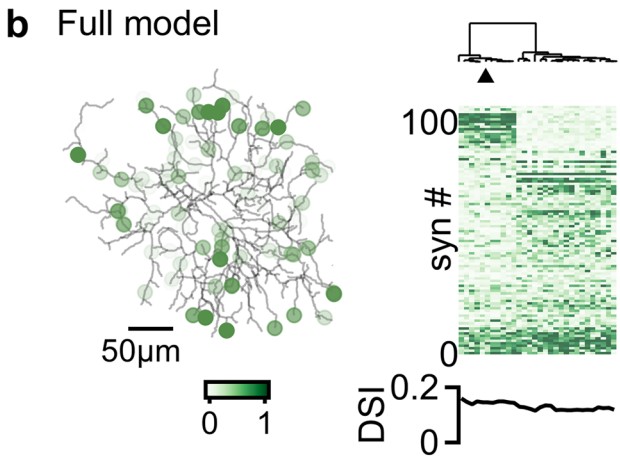

## b   Full model

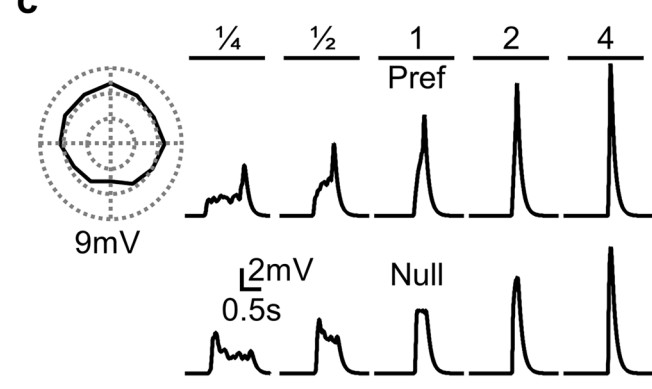

## c

**Fig. 3 | Directional computations in elementary motion circuits composed of identical presynaptic filters. a** Directional integration in the minimal model. The only source of asymmetry is in synaptic weights (left, highlighted in green). Detector responses (black) are a linear combination of the inputs (green). Inset, an alternative solution. For clarity, only the response of the presynaptic cell recruited first in the activation sequence is shown. **b** Left, representative optimal solution in the full model, color coding indicates synaptic weights. Right, hierarchical clustering of the null-preferred axis distribution of synaptic weights across simulation runs. Triangle, the model shown on the left and in (**c**). **c** Directional (left, speed = 1 mm/s) and velocity tuning of somatic voltage responses for the model highlighted in (**b**).

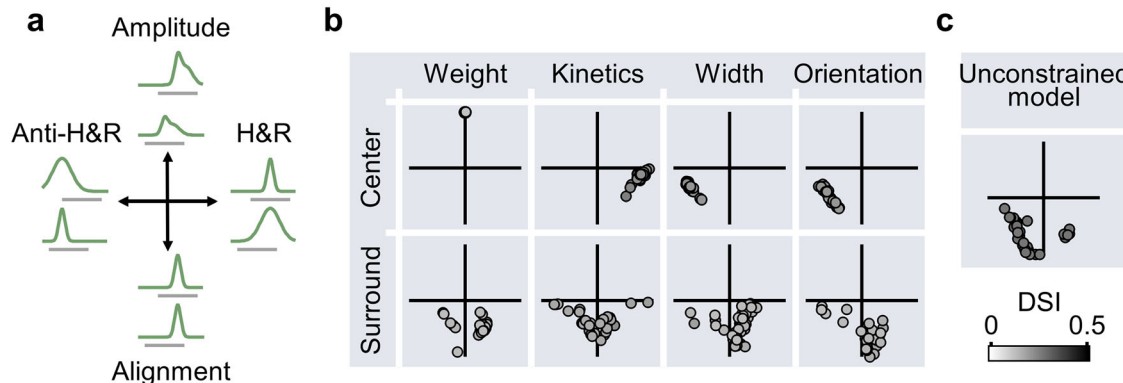

**Fig. 4 | Computational primitives in elementary motion detection circuits relying on excitatory inputs. a** Illustration of four distinct excitatory-based strategies for computing motion direction identified in this study. **b** The likelihood of observing the primitives in different circuit configurations. Cardinal axes correspond to the primitives illustrated in (**a**). Diagonal segments represent hybrid solutions that combine two distinct computational strategies. Color intensity reflects the DSI of each solution. **c** As in (**b**) for a model with presynaptic populations that were allowed to differ in all RF parameters.

synaptic weight or synaptic facilitation (Supplementary Fig. 3). But what algorithm produces the best directional performance? Removing all constraints on RF description illustrated that models can utilize diverse strategies to compute comparable DS levels (DSI = 27.8 ± 1%, Supplementary Fig. 4). Computational primitives provided a convenient platform for understanding signal processing in circuits with complicated filters. Across simulations, it was clear that the models evolved towards temporally aligning the responses in the preferred direction, either as a lone computational implementation or combined with H&R or anti-H&R kinetic differences, singling it as the best strategy to achieve the best selectivity without involving postsynaptic inhibition (Fig. 4c).

## Minimal requirements for DS mechanisms

The RF configurations described above differ substantially from the classical H&R model, which relies primarily on temporal filtering. This raises a question: what RF properties are truly required for DS computations?

In the formulation used here, each presynaptic RF contained two nonlinear components. First, synaptic dynamics incorporated depletion of the RRP, which altered both the gain and the shape of the input–output transformation. Second, because synaptic conductances were constrained to be non-negative, they imposed a rectifying nonlinearity on the visual signal.

To determine whether these nonlinearities were essential for DS, I again examined a minimal model containing only two presynaptic neurons and a linearly integrating postsynaptic detector. This simplified architecture retained the same RF definitions as the full multicompartmental model but was far easier to interpret. Using this framework, models in which center-surround RF kinetics, spatial extents, or relative amplitudes were allowed to vary continued to exhibit robust DS (Supplementary Fig. 5).

Replacing the nonlinear synaptic release mechanism with a purely linear formulation and removing signal rectification reduced, but did not eliminate, the DS produced by the spatial- and amplitude-based solutions when stimulated with moving bars or drifting gratings (Supplementary Figs. 5, 6). Thus, nonlinear RF interactions enhance, but are not strictly required for DS computations.

As shown above, center-surround interactions alone can generate DS. Indeed, spatial- and amplitude-varying models with linear synaptic release lost their DS capabilities when the surround component was not implemented (Supplementary Figs. 5, 6). Further analysis revealed that either nonlinear RF transformations or center–surround structure was sufficient to support direction discrimination for moving bars, whereas robust DS responses to drifting gratings required models in which the center and surround components were allowed to differ in temporal kinetics (Supplementary Figs. 5, 6).

## Enhanced performance in elementary motion processing models with a postsynaptic inhibitory drive

The models thus far examined circuit architectures where the detector receives innervation from populations of excitatory neurons. To study how postsynaptic inhibition affects directional processing in elementary motion circuits, I introduced an additional group of input cells that processed visual stimuli independently -- with the same organizational rules as described above -- and inhibited the postsynaptic cell. As before, I first tested whether this architecture could replicate known DS computations, in this case, the classic B&L formulation. The B&L model relies on spatially offset RFs, which can be achieved by allowing the presynaptic groups to have different synaptic weights[6]. Training models with variable strengths of the inhibitory drives achieved impressive directional performance (DSI = 50.8 ± 0.8%, Fig. 5a). The optimal solutions assigned a stronger inhibitory input on the null side of the dendritic arbor, leading to delayed recruitment of inhibition in the preferred direction, which is the essence of B&L computation (Fig. 5a)[37,46,75,76].

Notably, solutions to other functional circuit configurations did not always converge on B&L-like temporal interactions. Some models with kinetically diverse inhibitory cells, like the one shown in Fig. 5b, or with short-term plasticity of inhibitory synapses (Supplementary Fig. 7), were solved with an 'anti-B&L' arrangement, where early inhibition preceded delayed excitation in the preferred direction. In circuits with spatially varying inhibitory inputs, a different solution with a double-humped shape of the inhibitory profile that introduced a pause-in-inhibition around the time of the peak excitatory conductance was frequently observed (Fig. 5c, Supplementary Figs. 8, 9).

The information provided by the inhibitory drive can influence motion computations in ways that may not be reflected by the basic spatiotemporal filtering mediated by RF centers described above. Asymmetry in inhibitory RF surrounds can support impressive tuning levels (Fig. 5d–f, Supplementary Figs. 9, 10). Furthermore, while the DS performance of models with only the center or surround RF diameter was allowed to vary was rather low (Fig. 5g, h), it was significantly improved when both RF components of the inhibitory cells were variable (Fig. 5i, Supplementary Fig. 10).

As with excitatory DS circuits, I found four distinct computational primitives provided by inhibitory drive (Fig. 6). These include the classical B&L, 'anti-B&L', and 'pause-in-inhibition' mechanisms described above. Similar algorithmic solutions were observed in circuits evolved to produce DS responses to drifting gratings (Supplementary

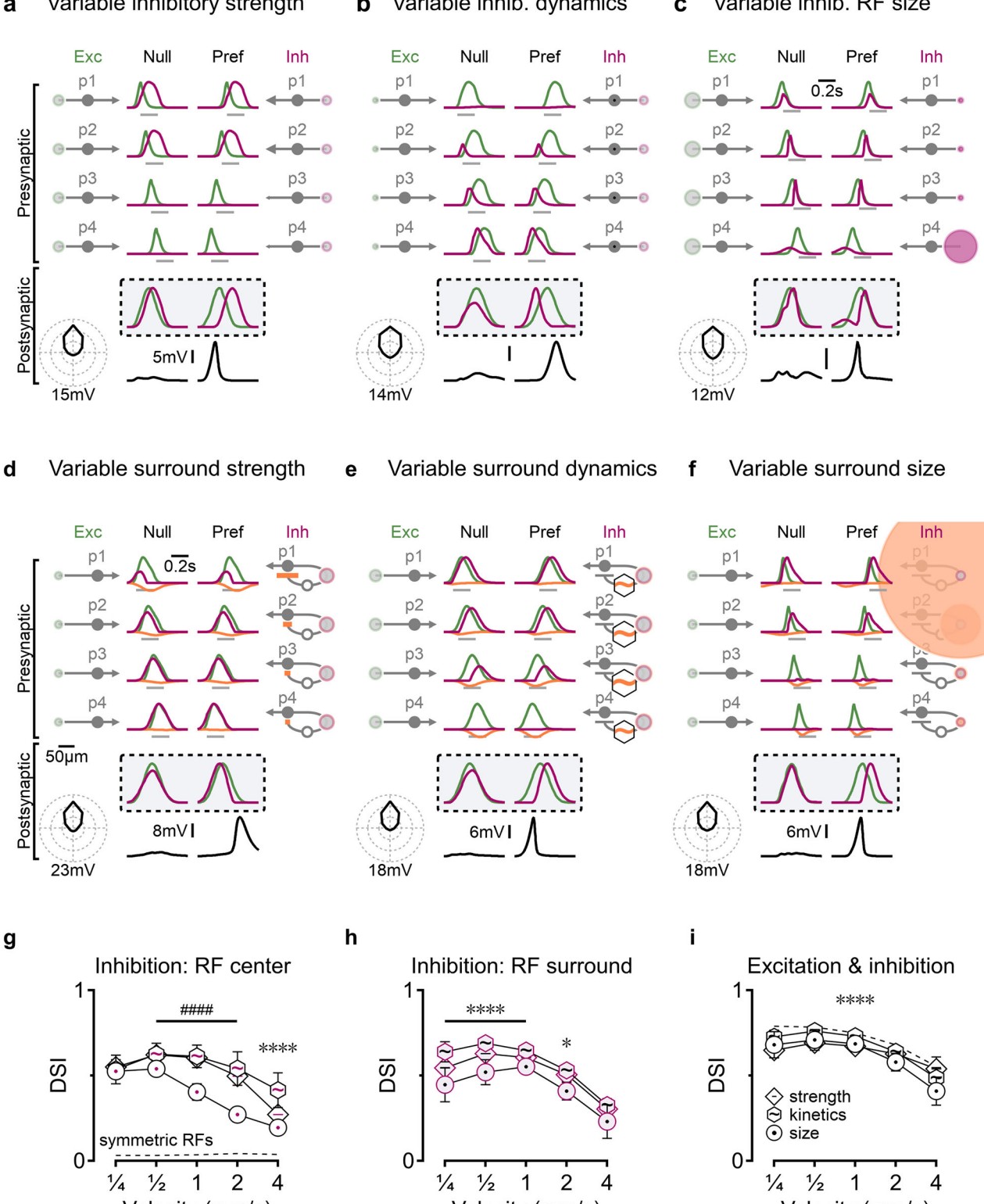

Fig. 11). Unexpectedly, while moving bar stimulation did not solicit solutions with directionally tuned inhibition, this computational primitive was observed as an optimal solution with drifting gratings (Supplementary Fig. 11).

More complicated interactions emerged when excitatory and inhibitory inputs were allowed to evolve. The two communication channels in this scenario supported a rich combinatorial space of possible interactions between various RF components. In general,

models with varying dynamics outperformed simulations with varying RF sizes or orientations: kinetic vs. spatial models reached peak DSI levels of $66.8 \pm 2.4\%$ vs. $56.2 \pm 8.2\%$ (Supplementary Figs. 9, 10). This enhanced performance of the temporal RF models could be attributed to more effective engagement of both excitatory and inhibitory mechanisms of DS, sometimes by combining H&R with B&L organization motifs (Supplementary Figs. 8–10)[14,77,78] or anti-H&R and anti-B&L mechanisms, with the latter often seen as the best performing

**Fig. 5 | Diverse computations observed in direction selective circuits incorporating postsynaptic inhibition. a** Representative solution observed in a circuit with constant excitation (green) and varying strengths of the inhibitory drive along the presynaptic input groups (magenta, presynaptic RF shown on the right). Inset, cumulative conductance onto the postsynaptic cell. Note the delayed inhibition in the preferred direction in accordance with the B&L model. **b** As in (**a**), but for a circuit with varying dynamics of the inhibitory inputs. In this solution, strong depolarization in the preferred direction was mediated by early inhibition that preceded the excitatory drive, indicative of an anti-B&L computation. **c** Pause-in-inhibition computation observed in a model with varying inhibitory RF sizes in which the inhibitory conductance is lowest when the excitatory conductance reaches its peak value in the preferred activation sequence. **d** A representative solution from models where the only variable parameter across inhibitory presynaptic groups was the strength of their surround receptive field. Surround components of the inhibitory inputs are shown in orange. **e**, **f** As in (**d**), but showing representative solutions from models in which the inhibitory surround groups differed in temporal dynamics (**e**) or spatial extent (**f**). **g** Summary of the mean velocity tuning in the three model families shown in (**a**–**c**). Dashed, a model with symmetric inputs. **h** Velocity tuning curves for model families depicted in (**d**–**f**). **i** Velocity tuning in models where both excitatory and inhibitory input groups were allowed to vary in strength, kinetics, and receptive field size. The dashed line shows the performance of an unconstrained model in which all receptive field parameters could vary freely. ****$p < 0.0001$, between different models. ####$p < 0.0001$, between variable inhibitory RF size solutions and other models, using ANOVA for each velocity with Bonferroni correction for multiple comparisons ($n = 100$ independently randomly seeded simulation runs). Error bars: standard deviation.

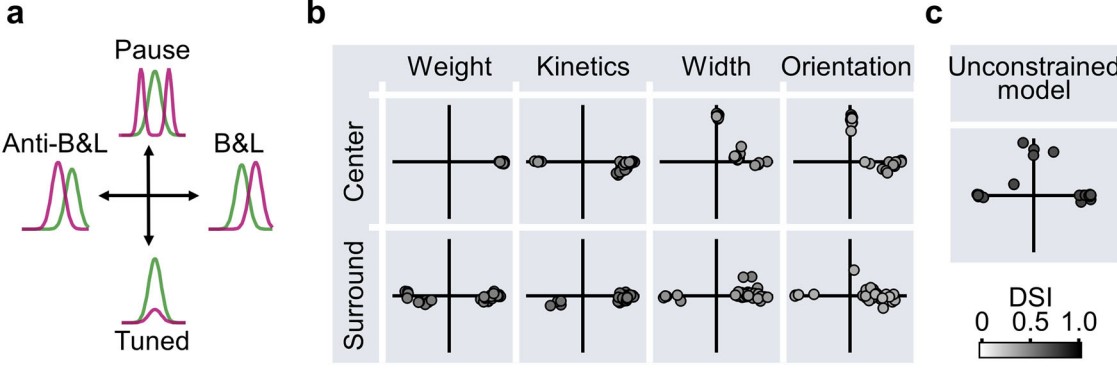

**Fig. 6 | Computational primitives supporting motion detection in circuits with inhibitory inputs. a** Schematic overview of the computational primitives in inhibitory-based DS mechanisms: directionally tuned inhibition, B&L, anti-B&L, and pause-in-inhibition. **b** Summary of DS computations across different inhibitory circuit configurations, as in Fig. 4b. Each cardinal segment highlights individual primitives shown in (**a**), while diagonal segments represent combinations of two distinct mechanisms. **c** Distribution of inhibitory computational primitives in a model where both excitation and inhibition could vary freely and independently.

solution in models unrestrained in all excitatory and inhibitory parameters (DSI = 73.1 ± 2.4%, Fig. 6c).

## Robustness of direction selectivity mechanisms across distinct neural architectures

To evaluate the generalizability of the computational principles identified in DSGC models, I applied the same machine learning-based exploration of DS to models of layer 2/3 cortical pyramidal neurons. The cortical system differs markedly from the retinal circuit in several key ways. First, synaptic input arises from spiking thalamo-cortical and cortico-cortical afferents, unlike the graded synaptic transmission characteristic of retinal circuits. Second, the spatial arrangement of synapses on the postsynaptic neuron does not reflect the retinotopy of their receptive fields[79,80]. Third, the asymmetric morphology of pyramidal cells, along with strong electrical filtering in apical dendrites, alters the integration of synaptic inputs and could influence motion computations[62].

Yet despite these structural and biophysical differences, the same DS mechanisms emerged in cortical models (Fig. 7). Together with similar findings in the minimal models (Fig. 3, Supplementary Figs. 3, 5), this result supports the conclusion that computational primitives guiding DS implementation described in this work are robust and largely independent of postsynaptic architecture.

## Robustness of direction selectivity mechanisms across variable stimulus presentations

For neuronal computations to be effective, they must remain robust in the face of noise or variability in sensory input. Previous studies have identified both population-level and cell-intrinsic mechanisms that support noise resilience in retinal and cortical circuits involved in direction and orientation selectivity[43,81–86].

To assess the noise tolerance of the different models described above, I evaluated their direction tuning in response to moving bars with variable acceleration. In each simulation run, the bar moved consistently in one direction, but its velocity fluctuated according to the noise level (Fig. 8a). This design ensured that performance reflected the circuit's ability to detect motion in the preferred direction, independent of confounding effects from motion in the opposite direction, as would occur if the velocity was negative. Thus, any degradation in directional tuning directly indicated reduced resilience to temporal variability within the stimulus.

As expected, increasing velocity variability led to a reduction in the DSI levels across most models (Fig. 8, Supplementary Fig. 12). Notably, two exceptions were seen. First, DS computation implemented with varying sizes of the center RF component performed exceptionally well and maintained high DSI values across all noise levels, in both excitatory and inhibitory setups (Fig. 8b–e). In contrast, models relying on variations in the size of the surround RF component performed poorly; their selectivity deteriorated rapidly with increasing noise, becoming virtually undetectable at higher variability levels (Supplementary Fig. 12).

## Methods to identify circuit components participating in DS computations

The conventional approach to identifying presynaptic mechanisms underlying DS typically involves whole-cell recordings or measurements of extracellular spikes or calcium signals. Most studies using this approach have employed stimuli designed to reveal temporal differences in input kinetics or spatial offsets between excitation and inhibition -- strategies motivated by classical models such as the H&R correlator, the energy model, and the B&L framework (Fig. 9a, b)[5–7]. Simulations of models trained to implement these computational

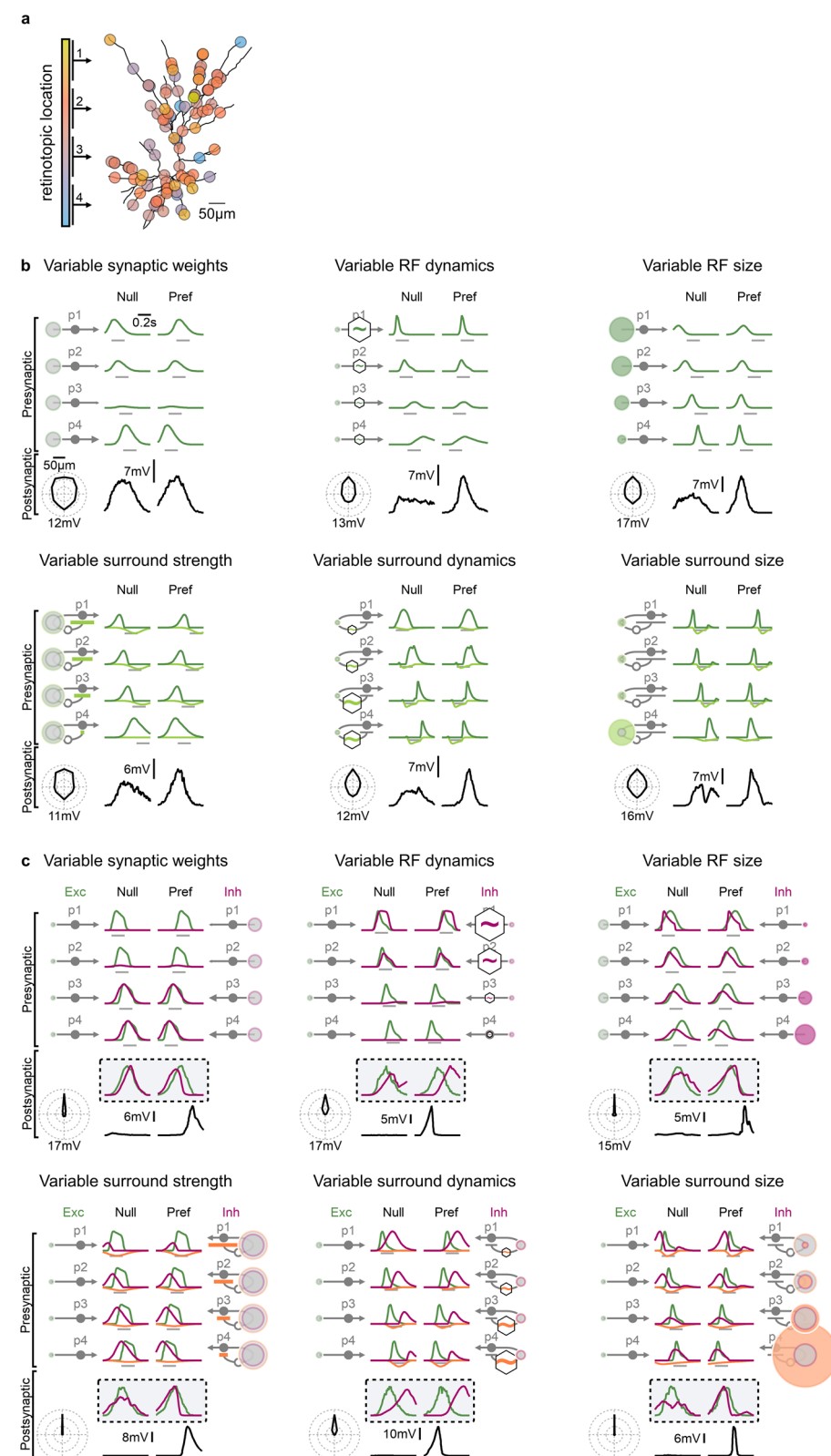

**Fig. 7 | Cortical layer 2/3 pyramidal neuron models recapitulate direction-selective computational primitives identified in the retinal circuit. a** Schematic of synaptic input organization in a model of a cortical layer 2/3 pyramidal cell. Unlike the retinal DS circuit, synaptic receptive fields were randomly distributed across retinotopic space and activated by spiking presynaptic neurons. **b** Representative DS solutions from a model receiving exclusively excitatory presynaptic inputs. **c** As in (**b**), but in a model that includes inhibitory presynaptic inputs, demonstrating the same algorithmic solutions seen in the DSGC model.

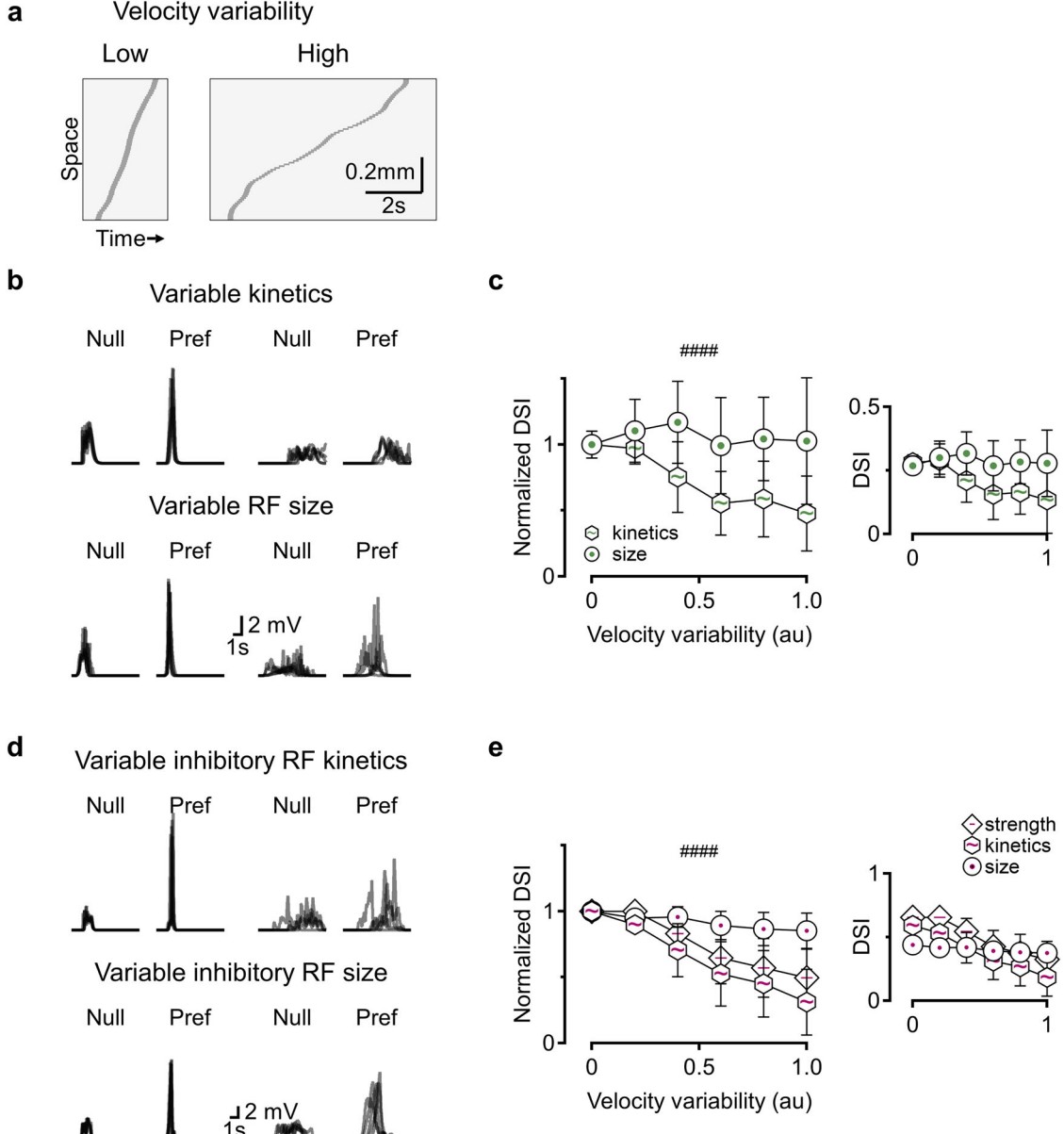

**Fig. 8 | Differential resilience of model solutions to stimulus variability. a** Space-time plots illustrating example trajectories of moving bars with speed updated every 50 ms, drawn from distributions of increasing variance. **b** Representative somatic membrane potentials recorded from DSGC models using either variable RF kinetics (top) or RF size (bottom) to compute DS, in response to stimuli with low (two left panels) or high (two right panels) velocity variability. **c** Summary of directional discrimination as a function of stimulus variability. The mean (±standard deviation) DSI values were normalized to the no-noise baseline (see inset for absolute DSI values). ####$p < 0.0001$; comparison of noise-dependence slopes using ANOVA on linear regression fits ($n = 100$ independently randomized simulation runs). **d**, **e** Same analysis as in (**b**, **c**), in which motion computations relied on inhibitory inputs with varied strength, kinetics, or RF size. ####$p < 0.0001$; significant difference between models using variable inhibitory RF size and other inhibitory configurations.

primitives confirm that these methods are effective at uncovering distinct spatiotemporal activation patterns, namely the presence of oriented space-time RFs, central to DS computation in those scenarios (Fig. 9c).

However, this traditional approach fails to identify DS circuits that rely on alternative mechanisms, such as spatial differences in receptive field structure or center-surround interactions (Fig. 9d–g, Supplementary Fig. 13). As a result, many of the DS-capable circuit configurations uncovered in this study remain inaccessible using conventional recording paradigms. Instead, these circuits require direct examination of the presynaptic population. Experiments targeting this layer -- via electrical recordings, calcium imaging of cell bodies, or measurements of synaptic activity (e.g., pre- or postsynaptic calcium signals or neurotransmitter release) -- could be combined with the same 1D noise paradigm to reveal asymmetries in center-surround RF structure among presynaptic cells (Fig. 9d–g, Supplementary Fig. 13). This strategy, combined with more sophisticated RF mapping from motion responses[49], could be used to uncover the broader range of computational primitives participating in motion detection proposed in this study.

## Discussion

A comprehensive and unbiased exploration of elementary motion-processing circuits yielded a strikingly large number of distinct functional organizations of a simple shallow feedforward network capable of extracting motion direction from sensory input. This diversity

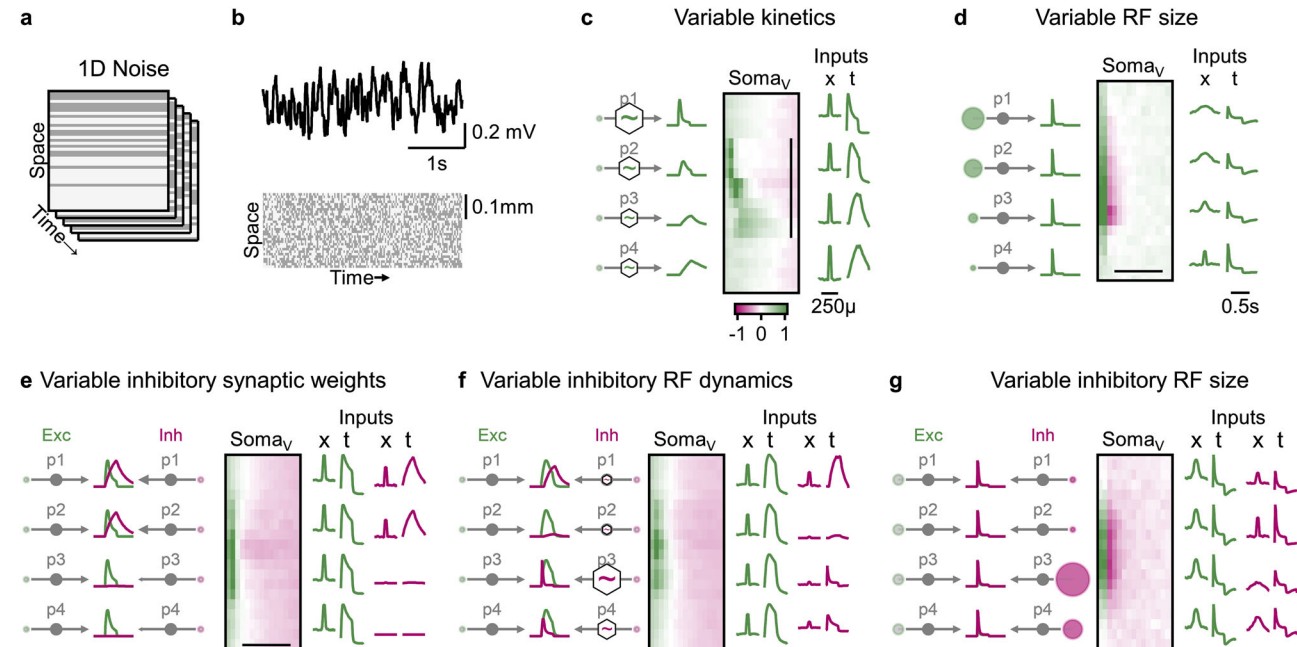

**Fig. 9 | Strategies for identifying diverse motion-processing circuits using 1D bar noise stimulation.** Receptive field structures were assessed using oriented bars aligned with the preferred direction of motion. **a** In each frame, individual bars were randomly assigned black or white contrast values to create a 1D bar noise stimulus. **b** Representative membrane potential trace recorded from the DSGC soma (top) in response to 1D bar noise stimulation (bottom) in (**a**) H&R model, where motion selectivity arises from differences in RF kinetics. **c** Left: Ground-truth spatiotemporal profiles of four excitatory presynaptic populations from an evolved model with variable RF kinetics. Temporal responses were derived from a full-field flash stimulus. Center: Space-time plot of the average postsynaptic (DSGC) response following the appearance of a white bar at each spatial location. The vertical axis indicates spatial position; the horizontal axis shows time after stimulus onset. Color coding represents depolarization (green) and hyperpolarization (red). Scale bar: 250 μm, centered over the DSGC's RF. Right: Inferred RF properties of the presynaptic populations based on their responses to the 1D noise stimulus. Both presynaptic and postsynaptic activity patterns reveal circuit organization. **d** As in (**c**), but for a model in which DS arises from differences in RF size. Scale bars: 500 ms. **e–g**, Analogous analyses for models in which motion selectivity is mediated by differences in inhibitory synaptic strength (**e**), RF dynamics (**f**), or RF size (**g**). In contrast to Hassenstein-Reichardt-type circuits, these implementations do not produce distinctive features in postsynaptic space-time maps, highlighting the need to examine presynaptic activity for circuit dissection.

emerged without imposing strong prior assumptions on network structure, demonstrating that direction selectivity can utilize multiple circuit asymmetries and arise through many alternative circuit configurations. Notably, some of these solutions resemble known biological implementations, including classical correlator models and feedforward inhibition schemes that have been observed in the retina, cortex, and insect visual systems. However, many other effective configurations uncovered here have not been described in biological tissue, likely because they lack a theoretical precedent and are not easily predicted by current models. By mapping out this previously unexplored space of viable circuit architectures, this work provides a theoretical framework that can now guide experimental efforts to search for these DS motifs in real neural systems.

This study intentionally focused on the most basic and tractable implementations of motion direction computation. Rather than attempting to model the full complexity of biological circuits or incorporate all known components of DS processing, I limited the scope to elementary circuit motifs. These foundational models capture the essence of motion detection using the minimal necessary components, enabling a focused investigation into the computational principles underlying direction selectivity. While this formulation may capture the key players in the bipolar-DSGC circuit, this simplification inherently omits some of the rich dynamics found in other biological systems. Yet it provides a crucial advantage: it allows for systematic exploration of how basic receptive field features and spatial arrangements give rise to directional tuning.

Even with the intentional reduction in complexity, fully enumerating and analyzing all potential circuit interactions within this constrained space proved challenging. Each model configuration -- defined by a set of spatiotemporal presynaptic filters, their spatial positions, and postsynaptic integration rules -- required independent testing to determine whether it could produce DS responses.

Notwithstanding these caveats, the advantage of the reduced model space is twofold. First, it facilitates a detailed and mechanistically transparent examination of the key receptive field components that can drive DS. Rather than relying on complex or biologically ambiguous nonlinearities, these models demonstrate how DS can arise from biologically-relevant input-output transformation and the spatial and temporal alignment of relatively simple synaptic inputs. For this reason, the models developed here are comparable in the complexity of circuit architecture to classic frameworks such as the full Hassenstein-Reichardt correlator or the energy model[5,7].

Second, the simplicity of these models makes them amenable to direct biological testing using standard physiological and anatomical techniques. Indeed, prior work already supports several of the DS mechanisms highlighted here. Li et al. (2018) reported spatial asymmetries in synaptic input strength and short-term plasticity onto DSGCs using whole-cell recordings. Intriguingly, analysis of bipolar-cell release onto ON-DSGCs, in addition to revealing faster temporal dynamics along the preferred-direction motion axis, reported expansion of the spatial RFs[38]. These measurements, obtained during routine RF characterization, align closely with predictions of the spatial RF model (Fig. 1). Complementary evidence comes from axonal calcium imaging: Hanson et al. (2023) identified spatially oriented RFs in type 5 A bipolar cells that innervate DSGCs, a presynaptic structure capable of contributing to DS computations. Beyond the retina, Lien and Scanziani (2018) found substantial variability in the spatiotemporal RFs of thalamic neurons projecting to cortical layer 4, although the

relationship between RF size and the preferred motion axis was not quantified. These studies demonstrate the feasibility of experimentally probing the RF features predicted here to support DS, including center–surround interactions and asymmetric spatial arrangements of presynaptic inputs (Fig. 9).

When analyzed systematically, functional implementations of direction selectivity in response to moving bars could be reduced to a small set of underlying computational primitives. These primitives capture the core operations (such as spatiotemporal asymmetry, timing of inhibition, or input strength) that enable motion detection, independent of the specific biological substrate in which they are embedded (Figs. 4, 6). Many of these primitives were observed in circuits evolved to process drifting grating stimuli. By distilling the full range of viable models into a compact set of algorithmic building blocks, this work offers a conceptual framework for understanding motion processing that transcends species or circuit-specific details. It also enables a more streamlined analysis of neural function: rather than grappling with the combinatorial complexity of synaptic configurations, one can now interpret DS circuits in terms of their membership in a limited set of computational strategies.

One known computational primitive is that DS could arise from directionally tuned inhibition impinging on the postsynaptic detector. However, when circuits were trained to respond to moving bars, directionally tuned inhibition did not emerge as the optimal solution for generating DS in the models examined (Fig. 6, Supplementary Figs. 7–10). The underlying limitation appears to stem from the models' inability to generate sufficiently strong direction tuning in either excitatory or inhibitory inputs when considered in isolation. In most cases, the difference in synaptic drive between preferred and null directions was too modest to support robust DS solely through directionally biased inhibition (Figs. 1–4).

It is important to emphasize, however, that this result does not imply that tuned inhibitory neurons lack a role in directional processing. Indeed, this algorithm was observed when the models were tasked with detecting the direction of motion of a sinusoidal grating (Supplementary Fig. 11), suggesting that tuned inhibition may not be the primary mechanism for generating DS within the framework of moving bars, but advantageous in other scenarios. In addition, in many biological visual systems, DS inhibition is observed, but it typically acts to refine or amplify directional tuning that has already been established by earlier stages of the circuit[46,76,87–92]. Thus, while directionally tuned inhibition may not be necessary for initiating DS in a minimal model, it plays a well-established role in modulating and stabilizing selective representations in downstream circuits.

Finally, applying machine learning approaches could enable the identification of computational modules with clearly defined functions, providing a framework to investigate how biological components integrate to drive brain computations. Supplementary Fig. 14 illustrates how the spatial RF solution identified in this work can be implemented in a neuromorphic device, resulting in a circuit architecture and components that are distinct from classical kinetic implementations.

In their thought-provoking work, Jonas and Kording highlighted the limitations of traditional neuroscience data analysis methods, showing that they fail to meaningfully describe information processing even in a simple artificial model organism, such as the Apple I microprocessor[93]. Recent advances in artificial intelligence have revitalized the field of neuroAI, offering powerful new tools to interpret both biological and man-made computational systems. Machine learning methods, as demonstrated here, go beyond statistical approaches by directly probing the logic of operations. While correlations in register occupancy fail to reveal the function of an adder in a microprocessor[93], machine-learning tools can identify such components and suggest novel algorithmic implementations of similar computations[94]. The flexible architectures of genetic algorithms, in particular, provide transparent, interpretable, scalable, and explainable insights into brain function. These AI-based

tools empower neuroscientists to investigate anatomically defined circuits, uncover mechanisms underlying specific computations, and intelligently design and analyze experimental data -- essential steps in our pursuit of understanding the brain.

## Methods

Multicompartmental simulations were performed using NEURON 8.2 (www.neuron.yale.edu/neuron). DSGC (352 segments) and cortical layer 2/3 (69 segments) morphologies were adapted from[49,83]. The initial global passive parameters were as follows: passive conductance $= 4e^{-4}$ S/cm$^2$, membrane capacitance $= 1\,\mu$F/cm$^2$, reversal potential $= -60$ mV and axial resistance $= 150\,\Omega$cm. Simulations were executed for 3–4 s with a time step of 0.1 ms.

The postsynaptic cells received innervation from 100 excitatory and potentially 100 inhibitory inputs, making one synapse each. Presynaptic responses and analysis were done in Igor Pro 9 (www.wavemetrics.com) or Python 3.10. In retinal configuration, presynaptic RFs were distributed randomly in a circle covering 300 μm centered on the soma of the DSGC and connected to the nearest postsynaptic dendrite. Cortical circuits covered a larger area (400 μm), and the connectivity was made to random postsynaptic branches.

To determine presynaptic RF activation, the spatial overlap between two-dimensional Gaussian functions describing the center and surround RF components and the shape of the stimulus (see below) was computed separately for each time step ($\Delta t = 1$–10 ms).

$Area_t$ (μm$^2$) corresponds to the normalized fraction of the stimulated RF area at time $t$, computed for each component. For full-field static flashes, the entire center and surround were activated when the stimulus was presented. For moving stimuli, $area_t$ was the pixel-by-pixel sum of the area of the Gaussian function describing the center or surround RF component that spatially overlapped with the stimulus (Eq. (1)):

$$area_t = \sum_x \sum_y stimulus_{x,y,t} \times e^{\left\{ -\left[ (x_\theta)/\frac{size_x}{\sqrt{\ln 2}} \right]^2 - \left[ (y_\theta)/\frac{size_y}{\sqrt{\ln 2}} \right]^2 \right\}} \tag{1}$$

Where $size$ (μm) is the full width at half maximum (FWHM) amplitude of the corresponding RF component. $x_\theta$ and $y_\theta$ represent centered and potentially rotated x and y pixel coordinates, calculated as (Eq. (2)):

$$x_\theta =$$
$$(x - center_x) \times \cos(\theta) + \left( y - center_y \right) \times \sin(\theta)\ y_\theta = -\left( x - center_x \right) \times \sin(\theta)$$
$$+ \left( y - center_y \right) \times \cos(\theta)$$
$$\tag{2}$$

Subsequently, RF center and surround responses at time $t$ were determined independently from the following equations (Eq. (3)):

$$RF_t = \left( area_t - RF_{t-1} \right) \times \Delta t / rise\,time + RF_{t-1} \times adaptation_{t-1}\ adaptation_t$$
$$= \max(0, adaptation_{t-1} - RF_t \times \Delta t / decay\,time)$$
$$\tag{3}$$

Where adaptation represents depletion of the RRP. In models where RRP dynamics were not simulated, the adaptation term was set to 1 for the entire duration of the simulation. Rise and decay times were bound to be larger than $\Delta t$. The widths were positive and were smaller than 200 μm for the center component.

The integration of center and surround RFs was calculated as the difference between the RF components' activation (Eq. (4)):

$$RF_{full,t} = \omega_{syn} \times max(0, RF_{center,t} - RF_{surround,t} \times \omega_{surround}) \tag{4}$$

Where $\omega_{syn}$ is the synaptic weight and $\omega_{surround}$ is the strength of the surround, bounded between 0 and 1. In retinal models, the postsynaptic synaptic conductance was set to $RF_{full}$ values. Synapses in

cortical circuits were activated by simulated presynaptic spikes, whose rates were generated based on $RF_{full}$ values: at each time point, a spike was generated if a random value drawn from a uniform distribution in the range [0, 0.3] was larger than the $RF_{full,t}$.

## Short-term plasticity

The amplitude of synaptic release was constant unless noted otherwise.

In simulations with short-term dynamics, short-term depression was implemented using a depression factor $syn_{dep,amp}$ in the range [0, 1]. With a starting value of 1, representing a full readily releasable pool, synaptic depression was modulated by RF activation at each time point by Eq. (5):

$$syn_{dep,t} = syn_{dep,t-1} \times syn_{fac,amp} \times \left(1 - syn_{dep,amp} \times RF_{full,t}\right) \quad (5)$$

Where $syn_{dep,t}$ is the depression factor at time $t$, which recovered with a time constant $syn_{dep,\tau}$ (ms) as follows (Eq. (6)):

$$syn_{dep,t} = syn_{dep,t} \times (1 - \frac{syn_{dep,t-1}}{syn_{fac,\tau}/\Delta t}) \quad (6)$$

To calculate the presynaptic release, $RF_{full,t}$ was multiplied by $syn_{dep,t}$.

Short-term facilitation was calculated using the following algorithm: facilitation value $syn_{fac,t}$ and steady-state facilitation of the releasable pool $syn_{fac,amp}$ were initially set to zero. At each time point, the value of $syn_{fac,t}$ was recalculated (Eq (7)):

$$syn_{fac,t} = syn_{fac,amp} \times RF_{full,t} + syn_{fac,t-1} \quad (7)$$

The facilitation value at each time point had a slow decay to zero with a time constant $syn_{fac,\tau}$ (units, ms; Eq. (8)):

$$syn_{fac,t} = syn_{fac,t} - syn_{fac,t} \times \Delta t / syn_{fac,\tau} \quad (8)$$

Last, presynaptic release was increased by a factor of $10^{syn_{fac,t}}$. As $syn_{fac,t}$ ranged from zero to one; facilitation could increase release amplitude by a factor of ten.

## Voltage-gated channels

To enable spiking responses, sodium and potassium conductances were introduced at the postsynaptic soma, following[95], with maximum densities of 10,000 and 2000 pS/μm², respectively. Spike counts were quantified as the number of voltage crossings at 0 mV.

## Training of genetic algorithms

For each generation, 10 NEURON models were executed simultaneously. In most simulations, the stimuli consisted of full-field bars moving at five different speeds: 0.25, 0.5, 1, 2 and 4 mm/s across 12 directions, presented within an 800 μm-wide square arena. The height of the bar was 1 mm, and its duration was 0.2 s. The intensity of the stimuli was set to 1, and the background was set to zero (AU).

In most models, the presynaptic population was divided into 4 groups. A separate RF description was instantiated with random values drawn from a uniform distribution for each of the presynaptic groups. In the synaptic weight model, one group with the same RF was used. The following limits to the initial values were used:

Response delay (only in models with kinetic changes), 0–50 ms. Center width FWHM, 10–200 μm. Surround width FWHM > 10 μm. Activation/inactivation/depression/facilitation time constants > 10 ms. Surround strength, 0–1. Depression/facilitation factors, 0–1.

In the first generation, random values were assigned to seed each network model. Likewise, random initial values were chosen for

passive conductance (ranging from 1e⁻⁵ to 1e⁻³ S/cm²), and axial resistance (constrained between 50 and 200 Ωcm).

In each model run, somatic membrane potential was recorded. For each stimulation speed, the DS index was calculated as a vector sum of the peak postsynaptic somatic potentials normalized by the scalar sum of responses (Eq. (9)):

$$DSI = \frac{\sum R_\theta \cos(\theta)}{\sum R_\theta} \quad (9)$$

Where $R_\theta$ is the response magnitude at direction θ, with θ = 0 set as the preferred direction. To reward generation of large postsynaptic responses, the following directional metric was computed (Eq. (10)):

$$directional\ metric = DSI \times \tanh(maxR_\theta) \quad (10)$$

Where $maxR_\theta$ was the largest response value across all directions of stimulation.

Subsequently, the models were ranked based on the average directional metric calculated over the five stimulation speeds. The best-performing model seeded the next generation of candidate solutions. Mutations were then applied to 9 models, leaving one intact to prevent search degeneration. For each parameter describing the presynaptic RF and postsynaptic properties, a normal distribution with a mean of 1 and a standard deviation of 10% was used to determine a scaling factor. This scaling factor was multiplied by the parameter value and combined with a random value drawn from a uniform distribution ranging from −0.015 to 0.015. Two mutation types were implemented: (1) Independent parameters, which were allowed to vary between input groups. (2) Shared parameters that mutated only in the first input group, and the new values were applied without change to other input groups.

Typically, the models were evolved over 300 generations. To ensure adequate search quality, low-performing models were evolved over 1000 generations. At a minimum, 50 independent runs were performed for each circuit configuration.

## Minimal models

Simplified DS models consisted of two presynaptic cells and a postsynaptic detector that summed presynaptic waveforms. Genetic algorithms trained the minimal models based on detector responses, with a reward function similar to the full model. The key difference was that the direction selectivity index was calculated from two directions and for a single velocity.

## Analysis

Amplitude-based DS mechanism was determined by analyzing the similarity of motion response waveforms. First, synaptic responses to a bar moving at 1 mm/s were temporally aligned to ensure independence from synaptic position. Next, response overlap was calculated for all time points where at least one of the synaptic waveforms exceeded zero. Overlap was considered absent if less than 75% of the waveforms reached 20% or less of their peak response. Otherwise, the overlap was computed using the average waveform across the synaptic population for that time point. Finally, the mean value of the overlap waveform was used as the metric for the amplitude-based mechanism.

The extent of the alignment-based DS mechanism was computed similarly, but using actual motion responses in the preferred direction and the overlap required 25% of the population waveforms to reach at least 20% or more of their peak responses.

The presence of H&R and anti-H&R was determined as follows: for each synaptic input, the duration of the motion response was calculated as the FWHM amplitude. A linear fit to the durations of synaptic responses determined if the responses were widening or narrowing along the preferred motion axis, corresponding to anti-H&R and H&R

mechanisms, respectively. Finally, the metric for the mechanism was calculated using the narrowest and widest responses as follows (Eq. (11)):

$$metric_{(anti)H\&R} = (1 - \frac{narrowest}{widest})^2 \qquad (11)$$

The position along the amplitude-alignment and (anti-H&R)-H&R axes was taken as the difference between the corresponding metrics.

To determine the inhibitory mechanism, two time points that divided the area under the curve of the excitatory response in the preferred direction into equal thirds were found. Anti-B&L and B&L mechanisms were computed from the fraction of the inhibitory drive in the same direction within the first and the last phase of the excitatory response, respectively.

The participation of the inhibitory amplitude-based algorithm was computed by Eq. (12):

$$metric_{amp, inhib} = \max(0, \frac{amp_{Null} - amp_{Pref}}{amp_{Null} + amp_{Pref}}) \qquad (12)$$

Where $amp_{Null}$ and $amp_{Pref}$ represent the peak amplitude of the inhibitory drive in the null and preferred directions, respectively.

The 'pause' mechanism was computed when both anti-B&L and B&L mechanisms were below 20% (Eq. (13)):

$$metric_{pause} = \frac{amp_{Null} - amp_{Pref^*}}{amp_{Null} + amp_{Pref^*}} \qquad (13)$$

Where $amp_{Null}$ was computed as before and $amp_{Pref^*}$ represents the peak amplitude of the inhibitory drive within the middle time window of the excitatory response.

Position along the cardinal axes was computed similarly to the excitatory mechanisms. A computational primitive was considered to be present if the position distance was more than 20% from the origin along the corresponding axis.

### Variable acceleration and 1D noise stimulation

To evaluate the robustness of evolved models under variable stimulus conditions, I introduced motion noise by updating the bar's velocity every 50 ms. Each new speed was sampled from the positive half of a normal distribution. The width of the distribution was set by the imposed noise level, up to a standard deviation of 5 mm/s. These variable-velocity trajectories were then presented in the preferred and null directions to calculate single-trial DSI values. DSI statistics for each noise level were obtained by averaging across 50–100 trials.

For 1D noise stimulation, I used 20 μm-wide vertical bars that spanned the entire visual field. Bar contrast was randomly assigned as either 0% or 100% with equal probability. Contrast values were updated every 20–200 ms. To analyze receptive field characteristics, I examined responses at either the model detector or presynaptic cells, focusing exclusively on the ON (rising) response phase. Signal fluctuations following bar onset were averaged in 20 ms time bins for each spatial location. These measurements produced space-time response maps (Fig. 9), from which the temporal RF component was derived by analyzing the shape of the response over the central position, and the spatial RF was extracted along the different spatial positions at the time of the maximal response.

### Drifting gratings stimulation

In some simulations, the models were trained to optimize responses to drifting sinusoidal gratings rather than moving bars. The gratings, presented at 100% contrast on a gray background, moved across the cell's receptive field at a speed of 1 mm/s. Five spatial frequencies were tested: 100, 200, 400, 800, and 1600 μm/cycle. DSI was calculated using the steady-state peak-to-peak voltage difference measured over the final 30% of each trial instead of the peak response amplitude.

### Reporting summary

Further information on research design is available in the Nature Portfolio Reporting Summary linked to this article.

## Data availability

Source Data are provided with this paper.

## Code availability

Simulation code can be found in the following repository: https://github.com/PolegPolskyLab/DS-mechanisms.

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

## Acknowledgments

I thank Drs. Benjamin Scholl and Gidon Felsen for their insightful comments on the manuscript. This work was supported by the National Institutes of Health (R01EY030841, R01EY035293).

## Author contributions

A.P.P. designed, performed, and analyzed computational modeling, secured funding and wrote the paper.

## Competing interests

The author declares no competing interests.
