## [Transparent Peer Review file · Nature Communications]

Machine Learning Discovers Numerous New Computational Principles Supporting Elementary Motion Detection

Corresponding Author: Dr Alon Poleg-Polsky

Version 1:

Reviewer comments:

Reviewer #1

(Remarks to the Author)

In this paper, Poleg-Polsky uses a genetic algorithm to find model parameters for circuits that are direction-selective, with connectivity constraints imposed by measured connectivity and neural morphology. The claim is that some of these motifs represent entirely new algorithms for direction-selectivity, beyond what has canonically been considered in the field: the dynamic differences associated Hassenstein-Reichardt correlators, Barlow-Levick veto mechanisms, and motion energy models. If this claim is correct, then this is an important and interesting paper.

I found this paper clearly written, and the models all looked like they performed as described. However, I had some fundamental questions about choices made by the author and how they affect the results. In some cases, as I describe below, I have doubts about whether some of these models should be considered direction-selective.

Major issues

1) The author uses maximum responses to assess DSI throughout. However, I'm unaware of any existing or proposed biophysical mechanism that would measure the maximum response of a neuron over time in order to make such a comparison. If the stimulus is linearly filtered, then a nonlinearity is required to generate a direction-selective mean signal. Taking the maximum is one option, but an awkward one to implement. The motion energy model supposes one could square (or rectify and square) a signal, then average it, in order to compare to signals – that is far more plausible. The difference is not just academic; I'm including a short snippet of Matlab code at the end of this review that shows how such a difference in nonlinearity can invert the direction-selectivity of a system. I think this work needs to carefully assess the impact of the chosen nonlinearity on its results.

2) I found the equations excessively difficult to read, in part because they used words instead of variables, but also because there were unit issues that I could not get past. (Equation and line numbers throughout would be nice to have.) In equation 1, for instance, $area_t$ seems to have a numerator in the exponent in which time has a unitless number ($length/(speed*t)$) subtracted from it. Units should definitely match here. Similarly RF_t in equation 2 shows that RF_t is equal to a set of terms including one that is $RF_{t-1}/(rise\ time)$, which again, has units that do not work out. I believe these are intended to be discrete versions of differential equations, but I don't think they are correctly written, and make it difficult to evaluate the operations being performed.

3) This may be the most important issue. When I was reading, I spent a lot of time thinking about the example in Figure 1c, which shows how identical temporal processing but increasing receptive field sizes results in directional responses to the moving bar (using the max of the trace). In the discussion, the author says that a minimal model with only linear filtering and the $max()$ nonlinearity reproduces this result and the others in Figure 1, Figure 2, and S2. My intuition for this RF-size change model is that when the bar comes from the large side, its entry into the component RFs is staggered in time, but when it comes from the small side, it hits small and large RF edges simultaneously, creating a higher max value when all the inputs are summed. If we rewrite this as a single spatiotemporal linear summation, it will in its simplest form just have, for instance, a higher weight on one side than the other and bilobed temporal filters throughout (so that it responds primarily on entry, as in the figures in 1c). This proposed filter has the same property described here: larger max responses to a bar moving one way than the other.

a. However, the linear RF for the model with changing receptive field widths is spatiotemporally *separable* by virtue of the identical dynamics at all points in space. This means it can be written as $h(x)f(t)$. Adelson & Bergen require spatiotemporal inseparability for a reason: if the spatiotemporal filter is separable, one can convolve the spatial component first, then the

temporal component second, and then show that for drifting sinusoids, the results for one direction vs. the other is just a phase shift in output sinusoids of identical amplitude. That is, it is provably the case that such a linear filter cannot yield directional responses to a sinusoid. This is why there exists such a strong emphasis on oriented space-time filters to generate direction-selective responses.

b. However, I believe such an LN model does yield the proposed responses to a moving bar. If so, should we think of that as a direction-selective model, if it responds equally to drifting sinusoids in both directions? Drifting sinusoids are a standard tool in visual neuroscience, so why should one prefer a bar to a sinusoid (or vice versa) for evaluating direction-selectivity? How should we think of these direction-selective models if they're stimulus dependent, even for super simple stimuli? I view this as a fundamental issue for this paper – why optimize using moving bars if the results are not direction-selective to moving sinusoids?

c. (It may also be that the full model including all the synaptic dynamics and facilitation — not the simple LN model approximation referenced in the discussion — is direction-selective to drifting sinusoids. In this case, it is incumbent on the author to sort out which aspects of the model are required: for instance, it could be the rectification at the model synapse is important. But the claims seem as though they would be quite different from the current ones being made.)

d. This broad formulation becomes a question for many of the other proposed mechanisms that appear to be able to be approximated by a separable space-time filter.

More minor comments

1) DSI color scale in figure 4 isn't that useful. Maybe move it to the range involved? 0-0.3?

2) I think there are two prior papers doing similar work using the fly connectome that might be appropriate to cite here: Mano et al. 2021 and Lappalainen et al. 2024. With reference to bars vs. sinusoids, it's useful to note that these both use natural scene inputs to optimize their models.

3) I had a difficult time understanding the meaning of anti-HR and anti-BL.

4) Discussion sentence: "For this reason, the models developed here are simpler than classic frameworks such as the full Hassenstein-Reichardt correlator or the energy model (Adelson & Bergen, 1985; Hassenstein, 1956), both of which rely on more elaborate nonlinear computations and specialized spatiotemporal filtering". I don't think this sentence is even remotely true. This work relies on an unknown mechanism finding the maximum of a timetrace, relies on quite complicated synaptic facilitation and dynamics, and integrates over entire neurons using the NEURON simulator. The motion energy model relies on linear filtering and squaring, while the HR model relies on linear filtering and multiplying. In fact, all the extra nonlinearities in this paper make it rather hard to assess in some cases what is really causing direction-selectivity in order to compare it with these simple models (see major comment above).

5) The paper says that the optimized models were "Analyzed systematically", but this seems to mean "understood by the process of staring". It's fine to analyze this way, but is there a way to plot the model parameters or cluster them or show that there's really only 4 or 6 dimensions?

6) In the discussion, the author notes he did not observe DS inhibition in these models. But are the models rich enough to generate DS inhibition? That would require a nonlinearity upstream of the inhibitory input to the cell. It was hard to tell whether this was in principle possible. It is also a bit surprising, considering the well-known direction selectivity of the SACs upstream of DS-RGCs that confer some if not most of their DS properties.

7) Since this is an entirely computational project, the code should be made freely available -- I did not see any links to a repository.

Matlab code snippet:

```
% test of when max resp vs. other measure could lead to opposite
% results for directionality, using a linear model to make the response and different
% nonlinearities to evaluate strength

f0 = zeros(1,40);
f1 = f0; % filter at point 1
f2 = f0; % filter at point 2

f1(11:21) = 1; % mostly low
f1(11) = 2; % high at beginning
f2(6:16) = 1; % mostly low
f2(16) = 2; % high at end

figure;
subplot(2,1,1);
plot(f1); ylabel('f1');
subplot(2,1,2);
plot(f2); ylabel('f2');
xlabel('time step');

r_plus = f1 + circshift(f2,[0 5]); % this simulates two delta function inputs displaced in time and space
r_minus = circshift(f1,[0 5]) + f2; % and the opposite direction

figure;
subplot(2,1,1);
```

```

plot(r_plus); ylabel('direction plus');
subplot(2,1,2);
plot(r_minus); ylabel('direction minus');
xlabel('time step');

disp('max of the two:');
disp(['r_plus,r_minus = ' num2str(max(r_plus)) ', ' num2str(max(r_minus))]);
disp('sum square of two:');
disp(['r_plus,r_minus = ' num2str(sum(r_plus.^2)) ', ' num2str(sum(r_minus.^2))]);

```

(Remarks on code availability)

I did not find any code referenced in the methods or available with the manuscript.

Reviewer #2

(Remarks to the Author)

In this manuscript, the author uses a machine learning approach to uncover algorithmic building blocks for direction selectivity in models of visual neurons. They find that, when optimized for direction selectivity, these models tend to fall into one of several computational “primitives,” some of which align with existing models of direction selectivity like the Hassenstein-Reichardt correlator or the Barlow/Levick model, which gives some credibility to the overall approach. Interestingly, this genetic algorithm approach also reveals some new mechanisms for direction selectivity that rely on things like differences in upstream receptive field sizes and orientations. Another key finding here is that downstream/postsynaptic integration schemes don’t much alter the basic upstream building blocks that the genetic algorithm learns, indicating that these mechanisms might be shared across different circuits like retina, cortex, and perhaps other visual systems. Overall, this is an interesting manuscript that has important implications for our understanding of direction selectivity. The genetic algorithm approach is very interesting and somewhat novel (or at least unusual), and it seems well suited for this question, although I do wish there were more details about the evolution of the models (see below). I have two primary concerns about the manuscript that I think need to be addressed. First, the details of the genetic algorithm and the evolution of the models was sparse, which made it hard to evaluate the results, in particular the claim about there being distinct computational “primitives.” Second, the narrow range of stimuli used makes it hard to evaluate how general these mechanisms are.

Major issues:

1. How are solution families / computational primitives defined?

Throughout the paper, it is not clear how distinct the model solutions are, and how often each type of solution was arrived at. How did you draw boundaries between solution families? The major concern here is that when looking at this huge diversity of models that emerge, you are imposing structure based on the expectations of what the primitives should look like, rather than the structure really being present.

- Can you show clustering for more of these solution families, as you do in Fig. 3?

- How did you classify a model in Fig. 4 as a “pure” model or a “hybrid” model if presumably a single simulation results in a model that belongs to one or another cluster?

- What fraction of the evolved networks came to each solution class? Does changing the hyperparameters of the genetic algorithm (mutation rate, etc) change this?

2. Stimulus dependence

It seems like all of these simulations were run with moving bars oriented perpendicular to the axis of motion. How dependent are the primitives on the identity of the stimulus? What if you used a thicker/large moving feature? Or even widefield motion? I don’t think you need to re-run every simulation but some exploration of different stimuli would bolster the claim that the model is learning computational primitives and not just strange solutions that only work for one stimulus (the gap-in-inhibition solution in Fig. 5 comes to mind).

3. Biological feasibility

The paper claims that these computational primitives are biologically feasible in pretty strong terms. E.g. the abstract “All mechanisms are biologically plausible and correspond to known physiological and anatomical motifs.” I suppose plausibility is subjective but throughout the results there are models that have a distinctly non-biological feel to them, like things with non-monotonic changes in RF size or orientation across the NP axis (Fig. 2, 5), or a single presynaptic cell with a huge RF (Fig. 6). Can this be addressed or softened?

4. Null / preferred nomenclature

The labeling of the “null side” and “preferred side” seems backwards in places. The text says the null side is the “position from which the null direction stimulus enters the RF” which is my understanding of the existing convention. But Fig. 1a has the null direction stimulus entering on what is labeled the “pref side.” And again in the description of Fig. 5 the text reads “... stronger inhibitory input on the preferred side ...” but again isn’t that the null side? That would agree with the general idea that the BL model relies on “null side” inhibition – a description included in, for example, the Fried 2002 paper cited there and I’m sure other papers as well. Can you clear up this discrepancy since it really confused me while reading the paper.

Minor

- Can you add a legend item for the dotted line in 1d (top)
- I'm confused about the description of the model in Fig. 3 – text says “band-like organization of stronger synaptic inputs along the motion preference axis” but isn't the band perpendicular to the NP axis? Panel b looks like there are stronger inputs on both the N and P side of the RF – how is this similar to the 2 input model and how does it generate DS responses?
- Call out to Fig. 1g – there is no Fig. 1g. I assume this should only refer to Supp Fig. 1?

(Remarks on code availability)

I don't see any code in the supplementary materials. I cannot evaluate the code.

Reviewer #3

(Remarks to the Author)

This is an important and timely manuscript that systematizes different computational approaches to motion selectivity. The manuscript is well written and technically correct. It was interesting to read how motion selectivity can be achieved not only through spatial differences in temporal filtering as in the HR detector but also through spatial differences in receptive field size, ratios, and surround properties. One aspect that I can suggest for improvement is that it was not clear how well clustering into different primitives worked, whether clusters were well separated or it was just a continuum between different axes (temporal kinetics, amplitude, receptive field size and asymmetry). In other words, if there is strong difference in orientation differences does it preclude observing differences in dynamics. Or to pose the question in yet another way: Figure 4B shows the distribution of different possible effects individually but what about correlations between them (combinations of primitives).

(Remarks on code availability)

Reviewer #4

(Remarks to the Author)

I appreciate the effort the author has made in exploring computational principles underlying elementary motion detection. The manuscript is ambitious in scope, presenting a wide range of simulated circuit architectures and proposing that these converge on a small number of “computational primitives” for motion detection. The use of minimal, tractable feedforward models as a way to isolate and examine core features of direction selectivity is a commendable strategy, and I agree with your stated rationale that such simplification can reveal useful insights.

However, despite the technical breadth of the work, I find that the manuscript does not convincingly achieve its conceptual goals, nor does it provide sufficient clarity about the key questions it is addressing. My concerns center around the overall framing and purpose of the study, rather than any specific technical errors in the simulations. Below, I outline several major issues that I believe need to be addressed for the study to be considered suitable for publication.

1. Lack of a Clear Scientific Question or Rationale

The manuscript surveys a large number of possible circuit configurations that can implement direction selectivity (DS), but it remains unclear what the main scientific question is. Are you testing specific hypotheses about biological mechanisms? Are you proposing new models for experimental validation? Or are you primarily interested in mapping the design space of motion-detecting circuits? This ambiguity makes it difficult to assess the contribution of the work.

You state in the introduction that the study aims to explore how basic receptive field features and spatial arrangements give rise to directional tuning. However, the simulations appear largely exploratory and descriptive, lacking a well-defined problem or hypothesis. Without a clear rationale, it is difficult to judge the value or relevance of the results beyond their existence.

2. The Study Feels Preliminary Rather Than Conclusive

Although the manuscript presents numerous models and simulation results, the overall impression is one of an exploratory analysis or early-stage investigation. Many of the findings—such as the rediscovery of Hassenstein-Reichardt (H&R) and Barlow-Levick (B&L)—like computations—are already well known and expected. The identification of new combinations or variants of these does not in itself constitute a major conceptual advance unless their biological plausibility or explanatory power is convincingly demonstrated.

The models are presented in a systematic but somewhat catalog-like fashion, and the paper often reads like a compilation of simulation experiments rather than a focused scientific narrative. This is not inherently a flaw, but I believe that, for a journal like *Nature Communications*, a stronger framing and clearer implications are expected.

3. Ambiguity of Machine Learning's Role

You describe using machine learning to explore a “vast combinatorial space” of circuit solutions. However, the exact nature of the machine learning method, its justification, and its biological relevance are not clearly explained. While evolutionary algorithms can certainly be used to optimize network parameters, the study does not make clear what has been learned from this process, other than the fact that many solutions exist.

Moreover, the manner in which models are selected and evaluated—based on performance metrics like directional selectivity index (DSI)—follows an ML benchmarking logic that is not necessarily meaningful in the context of biological interpretation. There is little discussion of how these models relate to known circuitry or why certain parameter regimes might be favored by real neural systems. This weakens the biological impact of the findings.

4. Biological Plausibility Is Not Adequately Addressed

Although you note that all mechanisms are “biologically plausible,” there is little evidence or discussion to support this claim. Many of the circuit motifs presented could just as easily be dismissed as mathematical possibilities with no relevance to actual neural systems. Without experimental data or stronger theoretical justification, it is impossible to tell which of these mechanisms the brain might actually use.

Furthermore, because the design space is enormous, the models explored here necessarily occupy a minuscule region of the total parameter space. This raises concerns about the representativeness of the results. If no constraints (biological, theoretical, or functional) are applied to restrict the search space, then the significance of any particular solution is unclear.

5. Lack of Hypothesis-Driven Structure

A key issue throughout the manuscript is the absence of hypotheses that can be tested or evaluated. This is particularly problematic given that many of the findings are not surprising: for instance, that asymmetric RFs or synaptic weights can produce DS. The absence of falsifiability or predictive power makes the work more descriptive than explanatory.

As you note in the discussion:

“I found a strikingly large number of distinct functional organizations of a simple feedforward network capable of extracting motion direction from sensory input.”

This sentence encapsulates the core problem: while the variety of possible implementations is acknowledged, the manuscript does not provide a framework for determining which of them are meaningful. The absence of such a framework reduces the scientific value of the result.

6. Presentation Style and Scope

The extensive number of figures (1–10), while individually clear, contributes to a sense of fragmentation. There is a lack of narrative cohesion tying the results together or building toward a central conclusion. The manuscript feels more like an organized summary of simulations than a focused argument.

The use of phrases like “previously unrecognized circuit motifs” or “new computational primitives” may overstate the novelty of what are, in essence, variations on known mechanisms. Without empirical grounding or theoretical analysis, the use of such terms may risk being misleading.

While the technical execution of the simulations is sound, and the general idea of using machine learning to explore motion detection mechanisms is potentially valuable, the current manuscript lacks the conceptual clarity and scientific focus needed to support its claims. I encourage the author to reconsider the framing of the work, tighten the narrative around specific questions, and critically assess the biological meaning of the results.

I hope this feedback is helpful in guiding the next steps for this promising but currently preliminary line of research.

Minor Comments

The manuscript lacks page and line numbers, which makes it difficult to reference or discuss specific passages in detail. Including these elements is important for facilitating constructive feedback and precise editorial communication.

There is inconsistency in figure panel labeling. For example, panels are labeled as “Ci, Cii” in the figures but referred to simply as “1C” in the main text. This unconventional and inconsistent notation can be confusing for readers and reviewers alike. Please consider standardizing the figure references to align with common scientific writing conventions.

(Remarks on code availability)

Version 2:

Reviewer comments:

Reviewer #1

(Remarks to the Author)

I thank the author for a clear set of answers to the reviewer questions, including mine. Many concerns of mine were addressed.

However, I'm still troubled by not understanding how the author's models become direction selective to drifting sinusoids when the inputs have nominally identical kinetic parameters, but different spatial parameters (in addition to the spatial offset). This is main point 3 and minor point 4 in the initial review and rebuttal. The models in question are the simple models Figure S2c. Here, the author agrees that “purely linear models” would not be able to be direction-selective, but then adds that the spatial models presented here as the simple models have sources of nonlinearity to drive direction-selectivity. (I'm glad we're on the same page about this fact about linear filtering.) Further, the author asserts that these models do not need to be spatiotemporally oriented and somehow generate direction selectivity in another way.

It seems reasonable to me that there could be nonlinearities in these models that generate the direction-selectivity where purely linear filtering (equal to the linear spatial and temporal filtering parameters in the model) would not. However, I don't

think it is sufficient to just say that there is some unidentified nonlinearity at play here. Since this spatially varying unit RF size is portrayed as counter to 40-odd years of dogma on oriented receptive fields, and since the author has complete knowledge of these models, I think it is incumbent on the author to narrow down exactly which nonlinearities are permitting this and how they effect processing to generate the direction-selectivity. (It looks as though the adaptation equation ReLU is the only nonlinearity in the photodetector equations upstream of the linear summation, if I'm understanding the simple model correctly.)

Concretely, I could easily imagine that this photodetector nonlinearity results in changes in kinetics when the receptive field is larger. If so, then this effect could rather be thought of as a parameterization problem—the model parameters seem to imply that the kinetics are the same for the two channels, but in reality they are not. Such a result would be consistent with the standard oriented RF interpretation; it would only mean that the orientation of the RF is obscured in the model's parameterization into linear and nonlinear components. If this is what's going on, then I do not think that this spatial RF size change constitutes a form of direction-selectivity distinct from the prior kinetic ones. Instead, it would be another manifestation of the same kinetic differences, but using recruited nonlinearities to arrive at the kinetic difference. There may be some other way that this model becomes DS; if so, and if the kinetics of the inputs to these stimuli truly are the same, then I do think the results could be framed as distinct from prior models.

Overall, I find the text in the manuscript a bit less insightful about these issues than the rebuttal. The front-end nonlinearities appear to be central to what's going on and this should be emphasized, and the nonlinearities explained with care, including their potential effects on kinetics when coupled to changes in spatial receptive field extents. Figure S13 is representative of the issue. The explanation makes only references to nominally linear processing in p1 and p2 here, referring to kinetics and spatial size. However, panel b cannot be direction-selective to sinusoids if all the processing is linear. Where are the nonlinearities? Which ones matter? Why do they get recruited more or differently for large than for small receptive field sizes? How do they generate the direction-selective responses in model S13b? I think these questions are critical in all the models where spatial receptive field differences are said to create DS signals without explicit parameter differences in the kinetics (inhibitory input strength, size, etc.). Are these changes in spatial receptive fields actually creating kinetic differences when the model is presented with sinusoids? If so, it's a completely different interpretation than saying that circuits may be DS by employing larger receptive fields on one side than the other.

One last point on this: the author seems to have refit the models to make them DS when presented with sinusoids. With the moving bar, an entirely linear system can be DS with the max-response metric. It might be informative to compare the two solutions to find out what nonlinearities were invoked in the sinusoidal case.

(Remarks on code availability)

Reviewer #2

(Remarks to the Author)

My major points 1, 2, and 4 were sufficiently addressed.

My point 3, about biological feasibility was dismissed with no changes to the manuscript that I can discern. I don't think I'm being unreasonable, and another reviewer also raised a concern about the claims of biological feasibility. To be clear, it isn't that I believe that diverse spatial and temporal tuning properties in upstream neurons is non-biological. But rather that in the present manuscript the DS that is observed in some models appears to rely on very specific, non-monotonic, or otherwise "strange" arrangement of those inputs. The author's response suggests that the particular arrangement doesn't matter, and that there are families of solutions that have different arrangements that work perfectly well. If this is the reason for the apparent non-biological flavor of some of the models presented in the main figures then I believe this should be at least demonstrated and explained to the reader, given the paper's strong claims about biological feasibility.

(Remarks on code availability)

I could not get main.py to run. I suggest making a requirements.txt file for installation, or making the repository an installable python package, and adding a readme with installation and basic entry point instructions.

Reviewer #3

(Remarks to the Author)

The manuscript has improved. Especially useful was the addition of comparison between moving bars and gratings. However, in the second reading of the manuscript, the work does strike as delivering new conceptual advance. My take-away message from the manuscript is that directional selectivity can be achieved in any system that possesses asymmetry in in receptive field shape and/or dynamics. If that is correct, perhaps it is useful to state this up front. I also agree with other reviewers concerns, especially those from Reviewer 4.

(Remarks on code availability)

Reviewer #4

(Remarks to the Author)

I appreciate the authors' careful and detailed responses to the initial review. Below I offer a few additional comments on

several key issues.

First, my earlier concern was not about the breadth of the topic per se. It is important not to conflate a “specific” research goal with a “narrow” one. A study can address a large, foundational question and still define a clear, specific objective; conversely, even a very narrow topic can be presented in a vague or diffuse manner.

Second, it remains unclear how the reported results can be meaningfully linked to actual brain mechanisms. What concrete experimental approaches could be used to explore such links, and what evidence supports the claim that the reported results are biologically plausible? Stating that the model is “constructed from known circuit components” does not by itself guarantee relevance to real neural systems. In a design that is not centered on testing a sharply defined hypothesis but instead on simulating a moderately realistic network and searching for interesting outcomes, it is difficult to know how specific modeling choices or constraints influence the results. Moreover, the model conditions used here — shallow architecture, random initialization, etc. — differs substantially from biological circuits, which makes strong claims about biological reality hard to accept.

Relatedly, the manuscript asserts that the work “reveals four new computational primitives distinct from the canonical H&R and B&L models.” Such predictions would indeed be exciting if the model could further lead to experimental observations in the brain. However, producing novel patterns from a partially biologically inspired network is always possible, and without stronger justification it is not clear that other investigators should be motivated to invest in experimental validation.

The authors summarize their approach as “both hypothesis-driven and predictive,” with the stated hypothesis that “DS is supported by implementations and algorithmic strategies beyond the canonical models.” As written, this hypothesis is so general that it is effectively unfalsifiable; it would be surprising if any reasonable dataset failed to support it. This makes the study appear less truly hypothesis-driven and more exploratory or post hoc in nature. The outcome — rediscovery of classical models and identification of new computational primitives — does not convincingly advance the field, as validation of classical models has been addressed extensively in prior work. Likewise, the prediction of “experimentally testable mechanisms” leaves open the key question of whether the newly proposed primitives are genuinely expected to exist and be observable in the brain, and if not, why not.

In summary, while the modeling framework is interesting, the manuscript would benefit greatly from a clearer articulation of specific, falsifiable hypotheses, stronger justification for the biological plausibility of the proposed mechanisms, and a more rigorous discussion of how the predictions could be tested experimentally.

(Remarks on code availability)

Version 3:

Reviewer comments:

Reviewer #1

(Remarks to the Author)

The author has done a fine and careful job addressing my comments. I especially appreciated Figs S4 and S5, which lay out clear where direction selectivity arises for sinusoids. I have only minor comments.

Minor comments

- 1) Overall, I wonder if there are a few places where this paper might benefit by adopting the terminology of separable/inseparable linear filtering and oriented space-time filter – the terminology from Adelson and Bergen that has been so helpful in that part of the field. I’m thinking especially of places like around line 186, where my reading is that you are saying (for the sinusoids) that you can create oriented space-time RFs by combining multiple inseparable space-time filters. (Though it’s more complicated because of the reLU on each c-s synaptic input, so the simple linear analysis of the motion energy model doesn’t necessarily hold.) For the simplified linear models, which are equivalent to the ME linear filtering step, this terminology might be especially apt.
- 2) Line 150: “anti-HR” is being defined as the neurons with longer responses on the null-direction side of the RF. But kinetics are identical, so the longer responses are stimulus dependent. This only looks “anti-HR” with bars as the stimulus. With sinusoids, it would not. Shouldn’t the listed properties be properties of the circuit, not the stimulus-circuit pairing?
- 3) Last, I remain a tad incredulous at the claim near line 404 that these numerical models are “simpler” than classical frameworks like the HRC and ME. The reasoning given is that the numerical models are simpler because each model configuration required independent testing. I cannot see how one follows from the other. I think there are many things you could say about the differences between these numerical models and the kind of analytically tractable models represented by the HRC and ME. The HRC and ME models are not biologically realistic, they are in fact overly simplified, so that the nonlinearities and phases and anti-symmetric subtractions are prescribed, rather than allowed to be fit parameters. For instance, the original HRC model, proposed in the 1956 paper, has two parameters: the distance between sensors and the time delay. It has a very restricted architecture to make up for the lack of parameters and to focus explicitly on the pairwise correlations that are the simplest hallmark of motion. However, I don’t think one can argue that it’s not simple. Overall, I think it would be worthwhile to spell out precise differences and how they contributed to the new findings, rather than arguing that

these numerical models (with nonlinear synaptic release dynamics and integrated neural morphology!) are somehow just 'simpler', which I think is hard to define, hard to justify, and an unnecessarily controversial claim to make.

Minor minor comments

- 1) Line 17 in abstract: "previously undescribed" instead of "newly discovered"?
- 2) Line 37: "computational motifs" rather than principles?
- 3) Line 79: "nonlinear RRP dynamics"? It might be worthwhile to emphasize linear and nonlinear operations throughout...
- 4) Line 115 and beyond: RF size gradient with identical positive only kinetics gets bar DS, but NOT sinusoidal DS. Perhaps some telegraphing of this later caveat is order?
- 5) Line 315: for HR+BL formulation, probably want to also cite Leong ... Clandinin 2016 JNeuro.
- 6) Line 386: caveat also: DS is for specific stimuli; paper now gives examples for detectors that work well for bars but not at all for sinusoids. This seems like a point to highlight. What stimuli should we all be using to measure DS? More of them?
- 7) Line 378: I didn't see a 'Second, ...' for enumerating clearly the second key insight.
- 8) Line 452: Author suggests that sinusoids are more complex than simpler bars. However, these are just two representations of space (regular and Fourier) and I'm not sure there's a good reason to claim that one is simpler or more complex than the other. If anything, most of the basic motion detection behaviors seem to be for detecting flow fields and stabilizing eyes and bodies – extended stimuli seem better suited for investigating/optimizing such detectors.
- 9) Line 456: Gruntman did not observe DS inhibition, since that does not appear to be a component of the fly DS circuit, so shouldn't be cited in this list.
- 10) Supp 14: In b, one could mention that this works as drawn for some stimuli (like bars) but not others, like sinusoids, unless you have spatiotemporally inseparable filters for p1 and p2? Consistent with Supp Fig 5.

(Remarks on code availability)

Reviewer #2

(Remarks to the Author)

All of my concerns have been addressed. Thanks for the thoughtful responses.

(Remarks on code availability)

Reviewer #3

(Remarks to the Author)

The author have adequately addressed the previous comments. The revised discussion clearly describes how the computational predictions made here can be tested in future studies, and how the existing experimental evidence on receptive field asymmetries provides early support for the predictions made here.

(Remarks on code availability)

Well, I have tried to install the code but it did not run on my machine. It is most likely a problem with my python environment. So, I cannot state that the code is not valid.

Reviewer #4

(Remarks to the Author)

The efforts made by the author to address the concerns raised by the reviewers are appreciated. While the revised manuscript and rebuttal may give the impression of engaging with these issues, many of the added explanations appear to stem from a misunderstanding of the underlying questions and are therefore largely irrelevant to the concerns raised.

As emphasized by other reviewers, the core issue remains the need to clearly establish the biological relevance of the findings — specifically, how the model's predictions relate to actual biological principles. While the model may produce simulation results that align with experimental data, this does not necessarily imply that it accurately reflects biological mechanisms, nor does it automatically confer scientific significance. Such significance typically arises from the formulation and validation of clearly falsifiable hypotheses, and this aspect appears to remain underdeveloped in the current manuscript.

(Remarks on code availability)

I appreciate the constructive feedback provided by the reviewers. Below you will find my responses to your concerns and suggestions marked in blue in this document and in the manuscript text.

I apologize for not including the link to simulation code in the main text, it appeared in the submission metadata: <https://github.com/PolegPolskyLab/DS-mechanisms>. The information is now included in the manuscript under 'code availability'

Reviewer #1 (Remarks to the Author):

In this paper, Poleg-Polsky uses a genetic algorithm to find model parameters for circuits that are direction-selective, with connectivity constraints imposed by measured connectivity and neural morphology. The claim is that some of these motifs represent entirely new algorithms for direction-selectivity, beyond what has canonically been considered in the field: the dynamic differences associated Hassenstein-Reichardt correlators, Barlow-Levick veto mechanisms, and motion energy models. If this claim is correct, then this is an important and interesting paper.

I found this paper clearly written, and the models all looked like they performed as described. However, I had some fundamental questions about choices made by the author and how they affect the results. In some cases, as I describe below, I have doubts about whether some of these models should be considered direction-selective.

Major issues

1) The author uses maximum responses to assess DSI throughout. However, I'm unaware of any existing or proposed biophysical mechanism that would measure the maximum response of a neuron over time in order to make such a comparison. If the stimulus is linearly filtered, then a nonlinearity is required to generate a direction-selective mean signal. Taking the maximum is one option, but an awkward one to implement. The motion energy model supposes one could square (or rectify and square) a signal, then average it, in order to compare to signals – that is far more plausible. The difference is not just academic; I'm including a short snippet of Matlab code at the end of this review that shows how such a difference in nonlinearity can invert the direction-selectivity of a system. I think this work needs to carefully assess the impact of the chosen nonlinearity on its results.

The reviewer is correct that peak membrane potential differences between trials do not always translate directly into proportional changes in spike output, as active properties in some neurons can rarely lead to complex nonlinear behaviors (see for example (Wienbar & Schwartz,

2022)). However, in both cortical and retinal direction-selective cells, stronger depolarization in the preferred direction is the norm, and this makes the voltage-based analysis biologically relevant.

The layout of the simulation closely follows classical experimental design and well-established analysis techniques, where the extent of direction selectivity is measured from responses to different trials in which stimuli move in different directions and the peak response or number of spikes are used to calculate the vector sum of the response, as I and others have done in the DSGCs. The following representative citations (too many to cite all!) from the cortical literature (Freeman, 2021; Lien & Scanziani, 2018; Wilson et al., 2018) and the fly DS system (Badwan et al., 2019; Gruntman et al., 2021; Maisak et al., 2013) show similar, peak-based analysis.

Please also note that not all DS cells spike, with one well-known example being starburst amacrine cells that generate directionally tuned calcium signals in their dendrites, but the analysis is quite similar (Euler et al., 2002; Morrie & Feller, 2018; Poleg-Polsky et al., 2018).

To further confirm the correspondence between subthreshold voltage signals and spiking output, I have now included simulations in which models were first trained on membrane potentials of the detector (the analog nature of the 'subthreshold' voltage signal has smooth derivatives, greatly simplifying model training) and then equipped post hoc with somatic voltage-gated sodium and potassium channels. As shown in the new Supplementary Figure 1, this approach yields robust, directionally tuned spiking responses that closely mirror the underlying voltage selectivity.

2) I found the equations excessively difficult to read, in part because they used words instead of variables, but also because there were unit issues that I could not get past. (Equation and line numbers throughout would be nice to have.) In equation 1, for instance, $area_t$ seems to have a numerator in the exponent in which time has a unitless number ($length/(speed*t)$) subtracted from it. Units should definitely match here. Similarly RF_t in equation 2 shows that RF_t is equal to a set of terms including one that is $RF_t/(rise\ time)$, which again, has units that do not work out. I believe these are intended to be discrete versions of differential equations, but I don't think they are correctly written, and make it difficult to evaluate the operations being performed.

Thank you for noticing. Units were added to all equations. I broke equation 1 into two components and now describe how rotation of elongated RFs was implemented. Old equation 2 is now fixed (it was missing a time step value)

3) This may be the most important issue. When I was reading, I spent a lot of time thinking about the example in Figure 1c, which shows how identical temporal processing but increasing receptive field sizes results in directional responses to the moving bar (using the max of the trace). In the discussion, the author says that a minimal model with only linear filtering and the max() nonlinearity reproduces this result and the others in Figure 1, Figure 2, and S2. My intuition for this RF-size change model is that when the bar comes from the large side, its entry into the component RFs is staggered in time, but when it comes from the small size, it hits small and large RF edges simultaneously, creating a higher max value when all the inputs are summed. If we rewrite this as a single spatiotemporal linear summation, it will in its simplest form just have, for instance, a higher weight on one side than the other and bilobed temporal filters throughout (so that it responds primarily on entry, as in the figures in 1c). This proposed filter has the same property described here: larger max responses to a bar moving one way than the other.

a. However, the linear RF for the model with changing receptive field widths is spatiotemporally *separable* by virtue of the identical dynamics at all points in space. This means it can be written as $h(x)f(t)$. Adelson & Bergen require spatiotemporal inseparability for a reason: if the spatiotemporal filter is separable, one can convolve the spatial component first, then the temporal component second, and then show that for drifting sinusoids, the results for one direction vs. the other is just a phase shift in output sinusoids of identical amplitude. That is, it is provably the case that such a linear filter cannot yield directional responses to a sinusoid. This is why there exists such a strong emphasis on oriented space-time filters to generate direction-selective responses.

I completely agree with the reviewer that exploring whether proposed mechanisms can respond to stimuli beyond a moving bar is important. As the new Supplementary figure 2 (recreated here) shows, both simple 2-input and the full models can create strong direction-selective responses to drifting sinusoidal gratings. This is true for models that vary in the spatial or temporal characteristics, and for models in which different manipulations of the surround were allowed, similar to figure 2. The revised manuscript now shows that all examined RF components are capable of responding preferentially to a specific direction of a drifting grating while utilizing the algorithmic solutions found with moving bar stimulation (new Supplementary figures 3 and 10).

Supplementary Figure 2. Responses of kinetic and spatial solutions to drifting grating stimulation. **a**, Top, space-time plot showing pixel intensity of a drifting grating stimulus activating two inputs positioned 200 μm apart. The model was trained to produce stronger activation (measured as the peak-to-trough amplitude during the steady-state phase) for motion from bottom to top. Schematic on left shows the temporal and spatial RF properties of the evolved model; horizontal line widths indicate RF size. Middle, stimulus intensity at the RF centers (gray) and response kinetics of the two cells (green). Bottom, linear summation of the two detectors (black). Vertical bars indicate response amplitude. Note the stronger signal in the preferred direction. Gray, linear combination of the stimulus intensity traces shows similar amplitude. **b**, Responses to five spatial frequencies in the full model with variable RF kinetics. **c-d**, As in **a-b**, but for models with variable RF diameters. In both cases, circuit architecture of the optimal solutions trained on drifting gratings were comparable to those trained on moving bars.

Reviewer's intuition is almost correct regarding the solution the models found for the spatially-diverse RFs. In the attached reviewer's figure 1, I illustrated the principle of such computation.

Figure R1. Toy examples of direction selectivity computation in models with spatially distinct receptive fields.

a, Schematic of two spatially offset cells with different RF sizes (red cell: large RF, blue cell: small RF; RF outlines shown as dotted curves). Soma positions are indicated by small circles. The RFs are positioned so that their edges meet at a single point. A narrow bar stimulus moves across the RFs, starting either from the side farthest from the meeting point (null direction, left, stimulus motion is from left to right) or from the opposite side (preferred direction, right, stimulus motion is from right to left). The spatial arrangement is depicted from the stimulus's point of view, so

temporal progression is left-to-right. Inset: same arrangement shown from an external observer's perspective, showing the actual direction of the stimulus (from right to left).

b, Model responses when each cell encodes stimulus appearance within its RF. In this toy example, the cell responds only when the stimulus reaches the cell. In the null direction, responses are temporally staggered, as the bar first enters the red cell's RF; in the preferred direction, responses are synchronous, enabling stronger summation. Vertical dotted lines mark stimulus entry into each RF.

c, DS computation in a scenario more representative of the manuscript's models, where each cell's input is proportional to the degree of overlap between the stimulus and its RF (dotted curves), followed by temporal filtering in both cells (solid curves). The temporal filter is identical in both cases, but because the overlap of the stimulus with the large RF sustains is longer, its filtered response is also prolonged.

d, Summed signal in a downstream linear detector depicting stronger activation in the preferred direction.

In principle, a model that responds to the entrance of the stimulus to the RF, can lead to DS detection that is speed invariant; in the arrangement shown in Fig. R1 a, the preferred-direction stimulus is always going to enter the RFs of both cells at the same time, drive responses and strong summation. Such a solution can be implemented with spiking cells, but I doubt if it would be able to operate over different contrasts, and I expect noise tolerance to be quite low.

A more realistic signal processing is shown in Fig. R1 b-c, where the cells' activation at any given time is a function of the overlap between the stimulus and the RF. Because this activation is fully symmetric for both directions (Fig. R1c, the arrangement of the dotted curves is a mirror version of each other), a temporal filter is required (Fig. R1c, solid curves). In this example, the filter is implemented with an adaptive synapse that mimics the depletion of the readily releasable pool (RRP). This two-step arrangement is conceptually similar to processing in bipolar cells, which sample (mostly) independent photoreceptor drives at their dendrites, which are summed and transferred as a membrane potential signal to the axon, where voltage-gated channels and synaptic dynamics perform the temporal filtering step.

Below is the code for recreating the model in Fig. R1 c-d (Igor syntax):

```
// preferred direction (x, y, last dim represents the two cells):
•make/o/n=(400,400,2) m_RF= (p-200+60*r)^2+(q-200)^2<(100-60*r)^2

// null direction
•make/o/n=(400,400,2) m_RF= (p-200-60*r)^2+(q-200)^2<(100-60*r)^2
```

```

// shared code

//convolve the stimulus with the RF (assume width=1 point)
•matrixop/o w_sum=sumrows(m_RF)
//normalize the stimulus (not strictly required for the model to work)
•matrixop/o w_max=maxcols(w_sum)
•w_sum/=w_max[0][0][r]
//this wave will hold the RRP availability
•make/o/n=(400,2) w_RF_adaptation=0
//RRP is decreased as a function of RF activation
•w_RF_adaptation[1,]=w_RF_adaptation[p-1][q]+w_sum[p][0][q]
//response profile, computed from the spatial RF activation and available RRP
•make/o/n=(400,2) w_RF_time=w_sum[p][0][q]*(1-w_RF_adaptation[p][q]/w_RF_adaptation[399][q])
//combination of the two cells' responses in downstream detector
•make/o/n=(400) w_RF_full= w_RF_time[p][0]+w_RF_time[p][1]

```

I took this detour to illuminate the difference between staggered responses and the solutions achieved with the genetic algorithm and presented in Figure 1. Importantly, the temporal envelopes of the signals that are transmitted to the detector during object motion are different between the presynaptic populations. Thus, although presynaptic cells in the spatial models (varying RF size and varying RF orientation) have similarly shaped responses to static stimuli, they are able to impose differential temporal filtering on moving stimuli, including drifting gratings. Thus, while the reviewer is correct that purely linear models would fail to respond to drifting gratings, the spatial models presented in the manuscript have sources of nonlinearity that allow them to have DS performance that matches the DS levels seen in kinetic models.

The reviewer raises an important difference between the kinetic solutions (Hassenstein & Reichardt and Adelson & Bergen models) and the spatial solutions discussed above. Kinetic solutions require oriented space-time filters; spatial solutions do not. I have mentioned this point in the manuscript in the results describing DS computations by the surround (Fig. 2d) and this was the motivation to show how these circuits can be identified experimentally (Fig. 10). Overall, this observation strengthens the claim that these are different solution families and some intuitions/requirements embedded in one solution family don't apply to the other.

b. However, I believe such an LN model does yield the proposed responses to a moving bar. If so, should we think of that as a direction-selective model, if it responds equally to drifting sinusoids in both directions? Drifting sinusoids are a standard tool in visual neuroscience, so why should one prefer a bar to a sinusoid (or vice versa) for evaluating direction-selectivity? How should we think of these direction-selective models if they're stimulus dependent, even for

super simple stimuli? I view this as a fundamental issue for this paper – why optimize using moving bars if the results are not direction-selective to moving sinusoids?

c. (It may also be that the full model including all the synaptic dynamics and facilitation — not the simple LN model approximation referenced in the discussion — is direction-selective to drifting sinusoids. In this case, it is incumbent on the author to sort out which aspects of the model are required: for instance, it could be the rectification at the model synapse is important. But the claims seem as though they would be quite different from the current ones being made.)

d. This broad formulation becomes a question for many of the other proposed mechanisms that appear to be able to be approximated by a separable space-time filter.

Again, thank you for the suggestion. As I now show, both the simple and the full models are directionally selective to drifting gratings. This provides further supporting evidence for the claim that the different new solutions discovered here (spatial, surround contribution, etc) are comparable in their ability to provide DS information to established kinetic solutions.

More minor comments

1) DSI color scale in figure 4 isn't that useful. Maybe move it to the range involved? 0-0.3?

Fixed

2) I think there are two prior papers doing similar work using the fly connectome that might be appropriate to cite here: Mano et al. 2021 and Lappalainen et al. 2024. With reference to bars vs. sinusoids, its useful to note that these both use natural scene inputs to optimize their models.

Thank you for the suggestion, references added!

3) I had a difficult time understanding the meaning of anti-HR and anti-BL.

My goal in this work is to analyze DS circuits at both the algorithmic and implementation levels. The literature contains many examples of different circuit implementations of established models, such as the H&R correlator. It is well known that multiple circuit configurations can realize the same computation. One of the questions I addressed was whether fundamentally new algorithmic solutions to the DS problem exist.

To explore this, it is first necessary to clearly define what constitutes a particular algorithm and what lies outside of it. In the H&R model, the core principle is the presence of distinct temporal filters arranged in a specific configuration: the slower (delayed) filter faces the preferred direction so that it is driven first. The same logic applies when detecting more complex stimuli, such as drifting gratings. Reversing the filter order in the H&R algorithm reverses the preferred direction.

However, in many spatially based solutions, the opposite occurs: the positions of the faster and slower filters are swapped, yet the detector still responds to the original preferred direction (Fig. 1). I refer to this computation as "anti-H&R."

Similarly, the essence of the Barlow–Levick algorithm is delayed inhibition in the preferred direction. In the "anti-B&L" solution, inhibition precedes excitation. In both the B&L and anti-B&L schemes, excitation and inhibition overlap in the null direction.

4) Discussion sentence: "For this reason, the models developed here are simpler than classic frameworks such as the full Hassenstein-Reichardt correlator or the energy model (Adelson & Bergen, 1985; Hassenstein, 1956), both of which rely on more elaborate nonlinear computations and specialized spatiotemporal filtering". I don't think this sentence is even remotely true. This work relies on an unknown mechanism finding the maximum of a timetrace, relies on quite complicated synaptic facilitation and dynamics, and integrates over entire neurons using the NEURON simulator. The motion energy model relies on linear filtering and squaring, while the HR model relies on linear filtering and multiplying. In fact, all the extra nonlinearities in this paper make it rather hard to assess in some cases what is really causing direction-selectivity in order to compare it with these simple models (see major comment above).

This is why I have included the simple 2 cell model with linear summation! All the key results from the full NEURON simulation were reproduced with just two inputs performing relatively simple, nearly linear, processing.

By comparison, the **full** Hassenstein–Reichardt model and the Adelson–Bergen (A&B) model are considerably more complex than many of the excitatory models presented in this manuscript. The full H&R model involves interactions between two DS detectors, while the A&B model incorporates separate ON and OFF channels.

5) The paper says that the optimized models were "Analyzed systematically", but this seems to mean "understood by the process of staring". It's fine to analyze this way, but is there a way to plot the model parameters or cluster them or show that there's really only 4 or 6 dimensions?

This is only partially true. I am a strong proponent of "staring" at the data (by which I mean exploring multiple types of visualizations to gain different perspectives on the underlying components). To this end, I dissected all models in a way that displays all major parameters. One such plot is reproduced below.

Figure R2. Illustration of components of the 'flexible surround size' model family.

Each row represents an optimal solution from the trained models. Columns, from left to right: Mean DSI measured across 5 stimulation velocities (tuning was assessed in 12 directions). Radar plot depicting the computational primitives recruited by each solution, color-coded by DSI. Excitatory conductance in the null and preferred directions, with strength represented by color intensity. The vertical axis represents spatial position over time. ω is the relative strength of the

surround for the four input populations. Activation and inactivation time constants are shown on a logarithmic scale. "Width" refers to the half-width at half-max amplitude of the corresponding RF component. The dendrogram shows the distance between solutions, calculated using hierarchical clustering. The models highlighted in Fig. 2c are indicated by black symbols. Note that the only parameter that varies between input populations is the width of the surround; other parameters are shared, but, of course, differ between specific implementations.

During this initial period of detailed inspection, I began to identify recurring computational motifs. In some solutions, input kinetics aligned with H&R predictions. Others showed similarly shaped response profiles, a configuration distinct from the H&R scheme, but these profiles were time-shifted to occur simultaneously in the preferred direction. As I show in Fig. 2d, this shift was absent for stationary stimuli. Still other solutions displayed different temporal shapes altogether, yet in positions inconsistent with predictions from kinetic models as discussed above.

After cataloging these patterns, I derived equations to distinguish between the different algorithms (see Methods for full details) and assigned names to the resulting computational primitives.

6) In the discussion, the author notes he did not observe DS inhibition in these models. But are the models rich enough to generate DS inhibition? That would require a nonlinearity upstream of the inhibitory input to the cell. It was hard to tell whether this was in principle possible. It is also a bit surprising, considering the well-known direction selectivity of the SACs upstream of DS-RGCs that confer some if not most of their DS properties.

As noted in the Discussion, the answer is both "yes" and "no." In our simulations, the presynaptic layer was constrained to have no DS responses. This was intentional: to (1) limit the parameter space (see discussion below), (2) adhere to the definition of an elementary motion-processing circuit with non-DS inputs, and (3) prevent the model from exploiting a trivial "shortcut" solution.

Nevertheless, the models could generate inhibitory DS responses in the postsynaptic cell using the same mechanisms that produced excitatory DS responses. However, even when inhibitory kinetics were allowed to vary, the models did not converge on H&R-like temporal profiles for inhibition. The reason was simple: other strategies consistently achieved stronger DS. This result changed when the models were optimized on the drifting grating task. As I show in the new Supplementary Figure 10, some models evolved to produce stronger inhibition in the null

direction. The Discussion now contains text that compares directionally tuned inhibition in bar and drifting grating stimulation .

7) Since this an entirely computational project, the code should be made freely available -- I did not see any links to a repository.

Matlab code snippet:

```
% test of when max resp vs. other measure could lead to opposite
% results for directionality, using a linear model to make the response and different
% nonlinearities to evaluate strength

f0 = zeros(1,40);
f1 = f0; % filter at point 1
f2 = f0; % filter at point 2

f1(11:21) = 1; % mostly low
f1(11) = 2; % high at beginning
f2(6:16) = 1; % mostly low
f2(16) = 2; % high at end

figure;
subplot(2,1,1);
plot(f1); ylabel('f1');
subplot(2,1,2);
plot(f2); ylabel('f2');
xlabel('time step');

r_plus = f1 + circshift(f2,[0 5]); % this simulates two delta function inputs displaced in time and
space
r_minus = circshift(f1,[0 5]) + f2; % and the opposite direction

figure;
subplot(2,1,1);
plot(r_plus); ylabel('direction plus');
subplot(2,1,2);
plot(r_minus); ylabel('direction minus');
```

```
xlabel('time step');
```

```
disp('max of the two:');
```

```
disp(['r_plus,r_minus = ' num2str(max(r_plus)) ', ' num2str(max(r_minus))]);
```

```
disp('sum square of two:');
```

```
disp(['r_plus,r_minus = ' num2str(sum(r_plus.^2)) ', ' num2str(sum(r_minus.^2))]);
```

Reviewer #1 (Remarks on code availability):

I did not find any code referenced in the methods or available with the manuscript.

Reviewer #2 (Remarks to the Author):

In this manuscript, the author uses a machine learning approach to uncover algorithmic building blocks for direction selectivity in models of visual neurons. They find that, when optimized for direction selectivity, these models tend to fall into one of several computational "primitives," some of which align with existing models of direction selectivity like the Hassenstein-Reichardt correlator or the Barlow/Levick model, which gives some credibility to the overall approach. Interestingly, this genetic algorithm approach also reveals some new mechanisms for direction selectivity that rely on things like differences in upstream receptive field sizes and orientations. Another key finding here is that downstream/postsynaptic integration schemes don't much alter the basic upstream building blocks that the genetic algorithm learns, indicating that these mechanisms might be shared across different circuits like retina, cortex, and perhaps other visual systems. Overall, this is an interesting manuscript that has important implications for our understanding of direction selectivity. The genetic algorithm approach is very interesting and somewhat novel (or at least unusual), and it seems well suited for this question, although I do wish there were more details about the evolution of the models (see below). I have two primary concerns about the manuscript that I think need to be addressed. First, the details of the genetic algorithm and the evolution of the models was sparse, which made it hard to evaluate the results, in particular the claim about there being distinct computational "primitives." Second, the narrow range of stimuli used makes it hard to evaluate how general these mechanisms are.

Major issues:

1. How are solution families / computational primitives defined?

Throughout the paper, it is not clear how distinct the model solutions are, and how often each type of solution was arrived at. How did you draw boundaries between solution families? The

major concern here is that when looking at this huge diversity of models that emerge, you are imposing structure based on the expectations of what the primitives should look like, rather than the structure really being present.

- Can you show clustering for more of these solution families, as you do in Fig. 3?
- How did you classify a model in Fig. 4 as a "pure" model or a "hybrid" model if presumably a single simulation results in a model that belongs to one or another cluster?
- What fraction of the evolved networks came to each solution class? Does changing the hyperparameters of the genetic algorithm (mutation rate, etc) change this?

Prompted by this comment, I now present the distribution of all individual solutions within the computational primitives space. This addition substantially increases the information available to the reader, revealing both the frequency of each primitive and their combinations.

The computational primitives form a continuum, quantified by equations provided in the Methods. A primitive was considered present if its positional displacement along the corresponding axis exceeded 20% of the origin. These quantitative definitions formalize the intuitions I developed by "staring" at the data (see response to Reviewer 1, comment #5). Importantly, every model exhibited at least one primitive (if this were not the case, the list of primitives would be incomplete). While additional algorithmic solutions to motion detection in these architectures may exist, they do not form stable local minima in the solution space. Indeed, some primitives emerged only under highly constrained parameter sets -- for example, the all-excitatory amplitude solution achieved only a modest DSI (Fig. 3). Models with flexible spatial or temporal RF properties generally did not optimize toward this primitive, as superior performance was attainable with H&R, anti-H&R, or alignment-based computations (Fig. 4).

Some models expressed more than one primitive. For instance, an all-excitation solution might show waveform width differences along the motion axis, but smaller than those in other implementations. According to my metrics, such a case would plot off the anti-H&R and H&R axes, leaning toward the "Alignment" mechanism pole (Fig. 4).

For clarity and more rigorous quantification, I divided the primitive space into eight octants: solutions in cardinal sectors were labeled "pure," while those on the diagonals were classified as "mixed."

As suggested by the reviewer, I now also show the clustering of all solution families in the corresponding supplementary figures.

Overall, the genetic-algorithm hyperparameters had little effect on the final outcome. As one would expect, a very low mutation rate could prevent convergence within a reasonable timeframe, while an excessively high rate would be counterproductive by stochastically pushing

the model away from stable gradient exploration. Across the tested range, mutation rates of 5–10% performed equally well, while 20% often failed. Populations of 10 models were as effective as 50, and most simulations converged to a reasonable solution in fewer than 100 generations, but were always continued to 300 generations to obtain the full potential of the solution.

2. Stimulus dependence

It seems like all of these simulations were run with moving bars oriented perpendicular to the axis of motion. How dependent are the primitives on the identity of the stimulus? What if you used a thicker/large moving feature? Or even widefield motion? I don't think you need to re-run every simulation but some exploration of different stimuli would bolster the claim that the model is learning computational primitives and not just strange solutions that only work for one stimulus (the gap-in-inhibition solution in Fig. 5 comes to mind).

I agree with the reviewer that this is an important benchmark for DS mechanisms. I have included exploration of responses to different bar widths and drifting sinusoidal grating (see detailed responses to reviewer 1). These simulations revealed a similar range of algorithmic solutions as observed with moving bars (see new Supplementary Figures 2, 3, 10). I believe this greatly strengthens the claim that these are, in fact, general-purpose DS solutions.

3. Biological feasibility

The paper claims that these computational primitives are biologically feasible in pretty strong terms. E.g. the abstract "All mechanisms are biologically plausible and correspond to known physiological and anatomical motifs." I suppose plausibility is subjective but throughout the results there are models that have a distinctly non-biological feel to them, like things with non-monotonic changes in RF size or orientation across the NP axis (Fig. 2, 5), or a single presynaptic cell with a huge RF (Fig. 6). Can this be addressed or softened?

The main reason for this effect can be attributed to model training. I did not attempt to generate monotonic function, nor were these required to replicate known solutions such as the

H&R implementation with kinetically – distinct inputs (see Fig. R3)

Figure R3. Different implementations of H&R detector with flexible kinetics solution families.

As in Figure R2, each row represents an optimal solution from the trained models, allowed to vary in input kinetics and lag time, indicating the offset between stimulus arrival to the cell and initiation of the response. Please note that some parameters change monotonically in some cases, but often have a prominent peak or trough indicative of slow / fast inputs, respectively, surrounded by members with less different dynamics.

Visual neuroscientists are well aware that presynaptic populations often include cells with widely differing kinetic properties, as illustrated above.

Likewise, neurons in the early visual system exhibit markedly diverse receptive field shapes and surround organizations. It is therefore reasonable to expect that circuit wiring could exploit these features to construct a directional detector. The minimal models demonstrate that asymmetry in these parameters, aligned with the preferred motion axis, can effectively promote direction selectivity. Importantly, some simulations converged on solutions where asymmetric cells were positioned away from the stimulus edge, underscoring the generality of the mechanism. This shows that the effect does not depend on highly tuned or contrived distributions -- a point I view as a strength, highlighting the biological plausibility of the solution rather than suggesting an artificial construct.

4. Null / preferred nomenclature

The labeling of the "null side" and "preferred side" seems backwards in places. The text says the null side is the "position from which the null direction stimulus enters the RF" which is my understanding of the existing convention. But Fig. 1a has the null direction stimulus entering on what is labeled the "pref side." And again in the description of Fig. 5 the text reads "...stronger inhibitory input on the preferred side ..." but again isn't that the null side? That would agree with the general idea that the BL model relies on "null side" inhibition – a description included in, for example, the Fried 2002 paper cited there and I'm sure other papers as well. Can you clear up this discrepancy since it really confused me while reading the paper.

Apologies for the confusion; the figures and the text were fixed with the existing convention.

Minor

- Can you add a legend item for the dotted line in 1d (top)

Added

- I'm confused about the description of the model in Fig. 3 – text says "band-like organization of stronger synaptic inputs along the motion preference axis" but isn't the band perpendicular to the NP axis? Panel b looks like there are stronger inputs on both the N and P side of the RF – how is this similar to the 2 input model and how does it generate DS responses?

Fixed 'perpendicular'. I have added the following text to the results section to explain and compare the band-like patterns:

“As in minimal models, two solution types were apparent. About half of the models evolved to have two bands of stronger synapses, with the band on the null side of the DSGC having more numerous inputs, mirroring the optimal solution seen in minimal models. In other cases, double strong-to-weak transitions evolved, each following the logic of the alternative minimal model solution (Fig. 3b, right).”

- Call out to Fig. 1g – there is no Fig. 1g. I assume this should only refer to Supp Fig. 1?

Fixed

Reviewer #2 (Remarks on code availability):

I don't see any code in the supplementary materials. I cannot evaluate the code.

Reviewer #3 (Remarks to the Author):

This is an important and timely manuscript that systematizes different computational approaches to motion selectivity. The manuscript is well written and technically correct. It was interesting to read how motion selectivity can be achieved not only through spatial differences in temporal filtering as in the HR detector but also through spatial differences in receptive field size, ratios, and surround properties. One aspect that I can suggest for improvement is that it was not clear how well clustering into different primitives worked, whether clusters were well separated or it was just a continuum between different axes (temporal kinetics, amplitude, receptive field size and asymmetry). In other words, if there is strong difference in orientation differences does it preclude observing differences in dynamics. Or to pose the question in yet another way: Figure 4B shows the distribution of different possible effects individually but what about correlations between them (combinations of primitives).

I appreciate the reviewer's suggestion and changed the computational primitives plots accordingly (reproduced below).

New Figure 4

New Figure 7

Indeed, in some instances, there is evidence of clusters, which tend to correspond to different circuit implementations mentioned in the main text. Excitation-only solutions (Fig. 4) were more likely to be a combination of two primitives (as indicated by models along the diagonals), whereas inhibitory outcomes were almost always restricted to a single mechanism.

Reviewer #4 (Remarks to the Author):

I appreciate the effort the author has made in exploring computational principles underlying elementary motion detection. The manuscript is ambitious in scope, presenting a wide range of simulated circuit architectures and proposing that these converge on a small number of "computational primitives" for motion detection. The use of minimal, tractable feedforward models as a way to isolate and examine core features of direction selectivity is a commendable strategy, and I agree with your stated rationale that such simplification can reveal useful

insights.

However, despite the technical breadth of the work, I find that the manuscript does not convincingly achieve its conceptual goals, nor does it provide sufficient clarity about the key questions it is addressing. My concerns center around the overall framing and purpose of the study, rather than any specific technical errors in the simulations. Below, I outline several major issues that I believe need to be addressed for the study to be considered suitable for publication.

1. Lack of a Clear Scientific Question or Rationale

The manuscript surveys a large number of possible circuit configurations that can implement direction selectivity (DS), but it remains unclear what the main scientific question is. Are you testing specific hypotheses about biological mechanisms? Are you proposing new models for experimental validation? Or are you primarily interested in mapping the design space of motion-detecting circuits? This ambiguity makes it difficult to assess the contribution of the work.

You state in the introduction that the study aims to explore how basic receptive field features and spatial arrangements give rise to directional tuning. However, the simulations appear largely exploratory and descriptive, lacking a well-defined problem or hypothesis. Without a clear rationale, it is difficult to judge the value or relevance of the results beyond their existence.

The main goal of this work is to identify the foundational principles that shape receptive field structures underlying direction selectivity. In the revised introduction, I now explicitly frame this guiding question. Understanding how receptive field properties influence motion processing is not an esoteric issue, but rather a central and still unresolved question in visual neuroscience. Hundreds, if not thousands, of studies have been dedicated to this topic. I believe the present work, building on this extensive foundation, contributes new and valuable insights.

The reviewer asks why the manuscript cannot be more narrowly focused. The reason is precisely because the question is so foundational. A narrow treatment would miss the broader implications. Addressing this problem requires not only theoretical models but also frameworks that can be tested experimentally, a systematic mapping of the design space of motion detection circuits, and consideration of whether these principles extend to applied domains such as neuromorphic engineering. Limiting the scope would risk reducing a fundamental inquiry to a case study, undermining the generality and impact of the work.

In general, theoretical studies are valuable because they can define broad organizing principles that unify mechanisms across systems and guide future experiments. For this reason, I believe a broad scope is not a weakness but a strength of the manuscript.

2. The Study Feels Preliminary Rather Than Conclusive

Although the manuscript presents numerous models and simulation results, the overall impression is one of an exploratory analysis or early-stage investigation. Many of the findings—such as the rediscovery of Hassenstein-Reichardt (H&R) and Barlow-Levick (B&L)—like computations—are already well known and expected. The identification of new combinations or variants of these does not in itself constitute a major conceptual advance unless their biological plausibility or explanatory power is convincingly demonstrated.

The models are presented in a systematic but somewhat catalog-like fashion, and the paper often reads like a compilation of simulation experiments rather than a focused scientific narrative. This is not inherently a flaw, but I believe that, for a journal like *Nature Communications*, a stronger framing and clearer implications are expected.

The "rediscovery" of the H&R and B&L mechanisms is not redundant but absolutely essential for the value of this work. There are two key reasons for this.

First, failure to identify these classical models would represent a major flaw. If the approach cannot recover well-established DS solutions, there would be no reason to trust it in uncovering novel ones. Demonstrating that the framework reliably rediscovers the canonical mechanisms provides a critical validation step that establishes confidence in its utility.

Second, and more importantly, the H&R and B&L models provide benchmarks for excitatory and inhibitory strategies of DS. In this project, I intentionally used both simplified motion detectors (linear integrations of two inputs) and elaborate multicompartmental DSGC models with realistic presynaptic input structure. In each case, the H&R and B&L solutions serve as reference points that illustrate how established DS mechanisms emerge in the two architectures.

By systematically altering single parameters in these models, I then showed that the same modeling framework can give rise to qualitatively different DS implementations. These new solutions are not trivial mixtures or variants of existing models. Rather, they reveal distinct algorithmic strategies for motion detection, ones that fall outside the canonical excitatory and inhibitory categories (see more discussion and examples below). The comparison to H&R and B&L therefore, does not dilute the novelty of the findings—it underscores it, by demonstrating that the new mechanisms emerge alongside but not within the established frameworks.

3. Ambiguity of Machine Learning's Role

You describe using machine learning to explore a "vast combinatorial space" of circuit solutions. However, the exact nature of the machine learning method, its justification, and its biological relevance are not clearly explained. While evolutionary algorithms can certainly be used to optimize network parameters, the study does not make clear what has been learned from this process, other than the fact that many solutions exist.

Moreover, the manner in which models are selected and evaluated—based on performance metrics like directional selectivity index (DSI)—follows an ML benchmarking logic that is not necessarily meaningful in the context of biological interpretation. There is little discussion of how these models relate to known circuitry or why certain parameter regimes might be favored by real neural systems. This weakens the biological impact of the findings.

I appreciate the reviewer's concern regarding the role of machine learning in this study. Let me clarify both the methodological rationale and the biological relevance.

First, the choice of an evolutionary algorithm is deliberate: this approach is well-suited for systematically exploring the combinatorial explosion of possible circuit configurations while remaining agnostic about specific mechanistic assumptions. Far from being a generic optimization exercise, this strategy allows one to uncover a range of solutions that traditional intuition- or hypothesis-driven modeling might miss. As I stated in response to previous concern, the rediscovery of canonical Hassenstein-Reichardt and Barlow-Levick mechanisms validates the method and demonstrates that it does not simply produce arbitrary solutions but instead converges on biologically meaningful architectures when they exist in the design space.

Second, the study is not claiming that "many solutions exist" as a trivial observation. Rather, it shows that distinct algorithmic strategies for motion detection emerge from small perturbations in model parameters, and that these solutions can diverge qualitatively from known models. This has direct biological significance: it suggests that direction selectivity can be implemented through multiple, functionally robust mechanisms, potentially explaining the diversity of DS circuits observed across species and cell types (see more below).

Third, the use of DSI here is not abstract benchmarking. With over 25 years of recording and analyzing single-cell responses in-vivo and in-vitro, I can easily identify realistic synaptic responses. In this work, I dedicated significant effort to finding the optimal optimization scheme that would be biologically relevant. I have resolved an ambiguity in the text to indicate that the ML did not optimize for a simple DSI metric, as the strategy often leads to very small responses (it is easier to have better DSI with small inputs due to the impact of the driving force on large EPSPs). The exact equation appears in the methods, and it contains two terms - DSI and postsynaptic response amplitude to force the models to produce strong postsynaptic

depolarizations. Overall, the fitness function rewards models that have strong and directionally tuned signals, which is biologically meaningful. Only after the training was complete, DSI was used to report on trained performance.

Why DSI? It is a standard and widely used measure of direction selectivity in experimental neuroscience. By using this metric, the models are evaluated on the same basis that real neurons are characterized.

Finally, the machine learning component is not an end in itself but a tool: 1) to validate the framework against classical solutions, 2) to discover qualitatively new DS mechanisms, and 3) to map parameter regimes onto biologically interpretable circuit features. Far from weakening the biological impact, this approach strengthens it by revealing principles that are generalizable, experimentally testable, and potentially explanatory of cross-species diversity in DS circuits.

4. Biological Plausibility Is Not Adequately Addressed

Although you note that all mechanisms are "biologically plausible," there is little evidence or discussion to support this claim. Many of the circuit motifs presented could just as easily be dismissed as mathematical possibilities with no relevance to actual neural systems. Without experimental data or stronger theoretical justification, it is impossible to tell which of these mechanisms the brain might actually use.

Furthermore, because the design space is enormous, the models explored here necessarily occupy a minuscule region of the total parameter space. This raises concerns about the representativeness of the results. If no constraints (biological, theoretical, or functional) are applied to restrict the search space, then the significance of any particular solution is unclear.

I respectfully disagree with the reviewer's assessment that biological plausibility is inadequately addressed. The study does not present arbitrary "mathematical possibilities," but instead is explicitly grounded in biologically relevant constraints at multiple levels.

First, the models are constructed from known circuit components of the retina / cortex and direction-selective cells. Inputs are drawn from realistic presynaptic populations, and the multicompartmental DSGC and L2/3 models incorporate experimentally established biophysical properties. The mechanisms that emerge are therefore not abstract mathematical constructs but solutions built on the same substrates available to real neurons.

Second, the evolutionary algorithm is not an unconstrained search through an infinite parameter space. The design space is intentionally bounded by biological considerations, such as synaptic weight ranges, short-term kinetics and receptive field structure, so that all solutions remain within physiologically reasonable regimes. This is precisely why the method successfully

rediscovers the canonical mechanisms: the constraints are sufficient to recover established biology.

Third, it is important to distinguish between circuit implementations (of which there can be many) and the underlying algorithms. These represent different levels of analysis, much in line with Marr's classic framework. At the computational level, the task is to detect stimulus motion. At the algorithmic level, the question is which computational principles can solve this task. At the implementation level, the focus is on how specific circuit "hardware" realizes those algorithms.

Going into this project, I anticipated that multiple distinct circuits could rearrange input filters in different ways to implement established algorithmic solutions. Demonstrating this is itself valuable, as it provides concrete hypotheses for how diverse neural circuits might arrive at similar functional outcomes and offers a foundation for future experimental testing.

The more significant finding, however, is that the known set of algorithmic solutions to direction selectivity is incomplete. This work reveals four new computational primitives that are distinct from each other and from the canonical H&R and B&L models. These results suggest that motion detection is not underpinned by a small, closed set of strategies, but rather by a broader algorithmic space that biological systems may draw upon in diverse ways. This suggests that DS can be implemented in multiple robust ways, which aligns with the empirical observation that different species and even different retinal circuits use diverse strategies for motion computation. The diversity of solutions is not a weakness but a biologically meaningful insight into the degeneracy and flexibility of neural circuit design.

Finally, the representativeness concern is directly addressed by the scope of the simulations. In the text, I avoided quantifying the parameter space directly because I think it is an unnecessary distraction. To give a sense of a single aspect of this space, below I show a schematic of different RF components that were examined in this study:

Figure R4. Exponential growth of model complexity with RF components.

Left: Illustration of some of the receptive field components considered in this work: both excitatory and inhibitory presynaptic cells can have center and surround components. Each can differ between presynaptic populations in four parameters: synaptic weight, response kinetics, spatial diameter, and degree of circularity. Configurations commonly analyzed in the literature are shown in gray. Right: Number of possible circuit architectures assuming each parameter can either be shared across or vary between presynaptic cells. This estimate was derived from a binary analysis, yielding: $N_{\text{architectures}} = 2^{\text{parameters}}$. For illustration, if only synaptic weight is considered, two architectures exist: uniform vs. variable weights. With two parameters, four combinations are possible: shared/shared, variable/shared, shared/variable, and variable/variable.

Note that in the manuscript, I also examined synaptic short-term plasticity, which adds another 2^4 possibilities. Clearly, testing all 2^{20} possible circuit configurations is infeasible within a single study. Instead, I focused on targeted perturbations of individual model components and organized the results in a roughly linear progression and examined questions that were not adequately addressed before, including:

1. How can spatial center RF characteristics impact motion computations in the absence of asymmetric kinetics?
2. Can RF surround mediate DS?
3. After RF properties are computed in the input population, can the amplitude and dynamics of signal transformation at the synapse to the detector affect DS?
4. How do responses to questions 1-3 change with an added inhibitory presynaptic population?
5. Are the newly discovered DS mechanisms relevant when the input is corrupted by noise?
6. How can they be detected experimentally?

Thus, while the combinatorial design space is indeed vast, the study systematically samples across it and compares outcomes against both canonical models and experimentally relevant benchmarks (e.g., DSI). This approach does not claim to exhaustively map every possibility but rather to identify the range of plausible solutions consistent with known biology and to highlight new mechanisms worthy of future experimental testing.

In short, the models are biologically grounded, the search space is meaningfully constrained, and the results provide both validation against classical mechanisms and discovery of novel, testable principles. Far from being arbitrary, the findings directly inform how neural systems may exploit circuit flexibility to achieve direction selectivity and provide a roadmap to uncover them experimentally.

5. Lack of Hypothesis-Driven Structure

A key issue throughout the manuscript is the absence of hypotheses that can be tested or evaluated. This is particularly problematic given that many of the findings are not surprising: for instance, that asymmetric RFs or synaptic weights can produce DS. The absence of falsifiability or predictive power makes the work more descriptive than explanatory.

As you note in the discussion:

"I found a strikingly large number of distinct functional organizations of a simple feedforward network capable of extracting motion direction from sensory input."

This sentence encapsulates the core problem: while the variety of possible implementations is acknowledged, the manuscript does not provide a framework for determining which of them are meaningful. The absence of such a framework reduces the scientific value of the result.

I again respectfully disagree with the characterization that this work lacks hypothesis-driven structure. The central hypothesis motivating the study is that direction selectivity can arise from a broader set of circuit implementations and algorithmic solutions than those currently recognized. This is a testable, falsifiable hypothesis: if the framework had recovered only the

classical models, then the conclusion would have been that no fundamentally new solutions exist or can't be uncovered using this approach. Instead, the discovery of additional computational primitives directly supports the hypothesis and expands the known solution space.

The reviewer is correct that certain findings, such as the fact that synaptic weights can generate DS, were reported before. These serve an essential role as positive controls.

The reviewer is concerned that the manuscript reports that asymmetric RFs can produce DS. But RF asymmetry is key! The existence of asymmetric RFs was not only known but is an absolute requirement. In fact, the motivating question for this work can be reformulated as “what are the asymmetric RF structures that can contribute to DS?” I find that essentially any asymmetry can be exploited by the model to generate DS responses. Many of such solutions are not biologically relevant - and I did not show them here. These include solutions with highly depolarized membrane potentials, models with extremely prolonged – seconds-long - responses, etc. But many more appear to be fully consistent with known physiology, and some are simply fascinating. To provide an example:

Many cortical cells are orientation selective. This classic property was discovered by Hubel and Wiesel more than 60 years ago, who showed how orientation selectivity arises from elongated RFs, which themselves can be created by a simple combination of circular center-surround input.

We know that many cortical cells can also produce DS responses. The current work proposes a simple organization scheme to convert inputs whose only asymmetry is in the orientation of their RFs to DS output at the detector. This result is conceptually as simple as the creation of oriented RFs by integration of presynaptic inputs from cells with displaced circular RFs, but as far as I know, this link between orientation and DS was not explored before. In the text, I mentioned recent work from the Awatramani group who found that the inputs to DSGCs can have oriented RFs (Hanson et al., 2023). Could these RFs contribute to DS? We do not know because the theoretical underpinning is missing and the key experiment – how are these oriented RFs organized as a function of the null-preferred axis was not performed.

For these reasons, far from being "merely descriptive," the work provides a structured framework for identifying, categorizing, and comparing algorithmic solutions to motion detection. The contribution is not only in enumerating many possible implementations, but in distilling them into algorithmic classes that reveal new organizing principles. This directly generates predictions that can be evaluated experimentally.

In short, the manuscript is both hypothesis-driven and predictive:

Hypothesis: DS is supported by implementations and algorithmic strategies beyond the canonical models.

Test: Apply an unbiased, biologically constrained search to see whether new solutions emerge.

Outcome: Rediscovery of classical models (validation) and identification of new computational primitives and demonstration how these could be realized in DS circuits.

Prediction: These circuits/primitives point to experimentally testable mechanisms and broaden the conceptual framework for DS.

Thus, rather than reducing the scientific value, the generality and hypothesis-generating structure of this work substantially increase its impact by providing both novel predictions and a principled foundation for future experiments.

6. Presentation Style and Scope

The extensive number of figures (1–10), while individually clear, contributes to a sense of fragmentation. There is a lack of narrative cohesion tying the results together or building toward a central conclusion. The manuscript feels more like an organized summary of simulations than a focused argument.

The use of phrases like "previously unrecognized circuit motifs" or "new computational primitives" may overstate the novelty of what are, in essence, variations on known mechanisms. Without empirical grounding or theoretical analysis, the use of such terms may risk being misleading.

I understand the reviewer's concern that the different solutions identified in this work might appear too similar, essentially variations on a common feedforward framework. It is true that all the models examined are feedforward circuits: they sample visual information through spatiotemporal receptive fields at the input layer and transmit the integrated signals to a postsynaptic detector via time-varying synaptic inputs. The key differences, however, lie in the parameters that define receptive field organization and, in some models, the kinetics of synaptic release. As elaborated above, these parameter choices give rise to qualitatively distinct mechanisms of direction selectivity, several of which are novel and not captured by classical formulations.

With that said, I appreciate the reviewer for raising this point, as it highlights the importance of clarifying these distinctions for readers who may not be experts in this area. In response, I have added a new supplementary figure (reproduced below) that directly contrasts temporal and spatial DS solutions, illustrating their distinct computational principles and demonstrating how each can be implemented in neuromorphic hardware. I hope this addition makes the

differences between the key implementations clear and strengthens the overall presentation of the work.

Supplementary Figure 13. Possible implementations of artificial motion detectors.

a, Schematic of a simple neuromorphic motion detector based on Hassenstein-Reichardt correlator logic. Two photodetectors (p1 and p2) feed into an analog integrator (adder). DS arises from differential temporal filtering: p2 is subjected to a delay or low-pass filter, producing stronger summation when motion engages the detectors in the upward direction. **b**, Similar schematic for a spatially based motion detector. Here, direction selectivity emerges from differences in photodetector size: p1 has a larger receptive surface area, while no temporal filter is applied. As demonstrated in this work, such spatial asymmetry can generate robust upward motion responses with selectivity comparable to the kinetic implementation.

While the technical execution of the simulations is sound, and the general idea of using machine learning to explore motion detection mechanisms is potentially valuable, the current manuscript lacks the conceptual clarity and scientific focus needed to support its claims. I encourage the author to reconsider the framing of the work, tighten the narrative around specific questions, and critically assess the biological meaning of the results.

I hope this feedback is helpful in guiding the next steps for this promising but currently preliminary line of research.

Minor Comments

The manuscript lacks page and line numbers, which makes it difficult to reference or discuss specific passages in detail. Including these elements is important for facilitating constructive feedback and precise editorial communication.

Fixed

There is inconsistency in figure panel labeling. For example, panels are labeled as "Ci, Cii" in the figures but referred to simply as "1C" in the main text. This unconventional and inconsistent notation can be confusing for readers and reviewers alike. Please consider standardizing the figure references to align with common scientific writing conventions.

Fixed

References:

- Badwan, B. A., Creamer, M. S., Zavatone-Veth, J. A., & Clark, D. A. (2019). Dynamic nonlinearities enable direction opponency in *Drosophila* elementary motion detectors. *Nat Neurosci*, 22(8), 1318-1326. <https://doi.org/10.1038/s41593-019-0443-y>
- Euler, T., Detwiler, P. B., & Denk, W. (2002). Directionally selective calcium signals in dendrites of starburst amacrine cells. *Nature*, 418(6900), 845-852. http://www.ncbi.nlm.nih.gov/entrez/query.fcgi?cmd=Retrieve&db=PubMed&dopt=Citation&list_uids=12192402
- Freeman, A. W. (2021). A Model for the Origin of Motion Direction Selectivity in Visual Cortex. *J Neurosci*, 41(1), 89-102. <https://doi.org/10.1523/JNEUROSCI.1362-20.2020>
- Gruntman, E., Reimers, P., Romani, S., & Reiser, M. B. (2021). Non-preferred contrast responses in the *Drosophila* motion pathways reveal a receptive field structure that explains a common visual illusion. *Curr Biol*. <https://doi.org/10.1016/j.cub.2021.09.072>
- Hanson, L., Ravi-Chander, P., Berson, D., & Awatramani, G. B. (2023). Hierarchical retinal computations rely on hybrid chemical-electrical signaling. *Cell Rep*, 42(2), 112030. <https://doi.org/10.1016/j.celrep.2023.112030>
- Lien, A. D., & Scanziani, M. (2018). Cortical direction selectivity emerges at convergence of thalamic synapses. *Nature*, 558(7708), 80-86. <https://doi.org/10.1038/s41586-018-0148-5>
- Maisak, M. S., Haag, J., Ammer, G., Serbe, E., Meier, M., Leonhardt, A., Schilling, T., Bahl, A., Rubin, G. M., Nern, A., Dickson, B. J., Reiff, D. F., Hopp, E., & Borst, A. (2013). A directional tuning map of *Drosophila* elementary motion detectors. *Nature*, 500(7461), 212-216. <https://doi.org/10.1038/nature12320>
- Morrie, R. D., & Feller, M. B. (2018). A Dense Starburst Plexus Is Critical for Generating Direction Selectivity. *Curr Biol*, 28(8), 1204-1212 e1205. <https://doi.org/10.1016/j.cub.2018.03.001>
- Poleg-Polsky, A., Ding, H., & Diamond, J. S. (2018). Functional Compartmentalization within Starburst Amacrine Cell Dendrites in the Retina. *Cell Rep*, 22(11), 2898-2908. <https://doi.org/10.1016/j.celrep.2018.02.064>
- Wienbar, S., & Schwartz, G. W. (2022). Differences in spike generation instead of synaptic inputs determine the feature selectivity of two retinal cell types. *Neuron*. <https://doi.org/10.1016/j.neuron.2022.04.012>

Wilson, D. E., Scholl, B., & Fitzpatrick, D. (2018). Differential tuning of excitation and inhibition shapes direction selectivity in ferret visual cortex. *Nature*, *560*(7716), 97-101. <https://doi.org/10.1038/s41586-018-0354-1>

Reviewer #1 (Remarks to the Author):

I thank the author for a clear set of answers to the reviewer questions, including mine. Many concerns of mine were addressed.

However, I'm still troubled by not understanding how the author's models become direction selective to drifting sinusoids when the inputs have nominally identical kinetic parameters, but different spatial parameters (in addition to the spatial offset). This is main point 3 and minor point 4 in the initial review and rebuttal. The models in question are the simple models Figure S2c. Here, the author agrees that "purely linear models" would not be able to be direction-selective, but then adds that the spatial models presented here as the simple models have sources of nonlinearity to drive direction-selectivity. (I'm glad we're on the same page about this fact about linear filtering.) Further, the author asserts that these models do not need to be spatiotemporally oriented and somehow generate direction selectivity in another way.

It seems reasonable to me that there could be nonlinearities in these models that generate the direction-selectivity where purely linear filtering (equal to the linear spatial and temporal filtering parameters in the model) would not. However, I don't think it is sufficient to just say that there is some unidentified nonlinearity at play here. Since this spatially varying unit RF size is portrayed as counter to 40-odd years of dogma on oriented receptive fields, and since the author has complete knowledge of these models, I think it is incumbent on the author to narrow down exactly which nonlinearities are permitting this and how they effect processing to generate the direction-selectivity. (It looks as though the adaptation equation ReLU is the only nonlinearity in the photodetector equations upstream of the linear summation, if I'm understanding the simple model correctly.)

Concretely, I could easily imagine that this photodetector nonlinearity results in changes in kinetics when the receptive field is larger. If so, then this effect could rather be thought of as a parameterization problem—the model parameters seem to imply that the kinetics are the same for the two channels, but in reality they are not. Such a result would be consistent with the standard oriented RF interpretation; it would only mean that the orientation of the RF is obscured in the model's parameterization into linear and nonlinear components. If this is what's going on, then I do not think that this spatial RF size change constitutes a form of direction-selectivity distinct from the prior kinetic ones. Instead, it would be another manifestation of the same kinetic differences, but using recruited nonlinearities to arrive at the kinetic difference. There may be some other way that this model becomes DS; if so, and if the kinetics of the inputs to these stimuli truly are the same, then I do think the results could be framed as distinct from prior models.

Overall, I find the text in the manuscript a bit less insightful about these issues than the rebuttal. The front-end nonlinearities appear to be central to what's going on and this should be emphasized, and the nonlinearities explained with care, including their potential effects on kinetics when coupled to changes in spatial receptive field extents. Figure S13 is representative of the issue. The explanation makes only references to nominally linear processing in p1 and p2 here, referring to kinetics and spatial size. However, panel b cannot be direction-selective to sinusoids if all the processing is linear. Where are the nonlinearities? Which ones matter? Why do they get recruited more or differently for large than for small receptive field sizes? How do they generate the direction-selective responses in model S13b? I think these questions are critical in all the models where spatial receptive field differences are said to create DS signals without explicit parameter differences in the kinetics (inhibitory input strength, size, etc.). Are these changes in spatial receptive fields actually creating kinetic differences when the model is presented with sinusoids? If so, it's a completely different interpretation than saying that circuits may be DS by employing larger receptive fields on one side than the other.

One last point on this: the author seems to have refit the models to make them DS when presented with sinusoids. With the moving bar, an entirely linear system can be DS with the max-response metric. It might be informative to compare the two solutions to find out what nonlinearities were invoked in the sinusoidal case.

I agree with the reviewer that clarifying how non-kinetic solutions generate motion enhancement for drifting sinusoidal gratings is essential. In response, I substantially revised the presentation of these solutions and now include a more explicit, mechanistic comparison of the different computational strategies that can produce direction selectivity in this regime.

Definitions used in this response.

For clarity, I refer to operations that preserve the sinusoidal form of the response (such as shifting, multiplication, or convolution) as linear. These operations keep each presynaptic response a sine wave (or a sum of sine waves). In contrast, processes that alter the waveform shape are termed nonlinear. For simplicity, I assume that the postsynaptic detector performs a simple linear summation of its presynaptic inputs, and I quantify DS as the difference in peak-to-peak amplitude of this summed postsynaptic response. For example, a fluctuation from -70 mV to -50 mV carries more directional signal than a fluctuation from -61 mV to -59 mV; various downstream mechanisms could further amplify this difference, but such steps are beyond the scope of this discussion.

Nonlinearities in the RF formulation

The model includes two sources of nonlinearity:

- Readily releasable pool (RRP) dynamics - synaptic release depletes the RRP, which in turn reduces subsequent release. This depletion-recovery loop is nonlinear.
- Center-surround rectification (ReLU-based operation that combines center and surround components). This nonlinearity is absent in models that use the center component alone.

In contrast, the photoreceptor drive onto each RF is integrated linearly: the RF template (a 2D Gaussian) is multiplied by the stimulus at each time step. As a result, drifting sinusoidal gratings always generate sinusoidal signals following spatial RF integration; smaller RFs generate larger-amplitude sinusoids, whereas larger RFs generate smaller-amplitude ones, but phase and frequency remain fixed.

This behavior contrasts with moving bars, where changes in RF size create differences in response kinetics. With a bar, small RFs experience a brief, transient activation while large RFs integrate the stimulus over a longer time window.

Solutions for drifting gratings

Do drifting gratings require nonlinear computations to achieve DS?

Not necessarily. Classical temporal models can generate DS purely through phase shifts. For example, presynaptic cells with fixed delays produce sine waves of identical frequency but offset phase. Linear summation of two such inputs with different delays is direction selective, a well-known solution that can be implemented compactly (e.g., code accompanying (Borst, 2018)).

In my model, RF size or amplitude alone cannot directly shift the temporal phase. However, linear solutions remain possible via center-surround interactions. When center and surround RFs are allowed to differ in their temporal activation timescales, their linear combination yields direction-dependent phase relationships, even without invoking the nonlinear components. These mechanisms are presented in the new Supplementary Figure 6 and described in the revised text. As seen in that figure, linear center-surround models achieved DSIs of ~15–25%, compared to ~50% when nonlinearities (RRP depletion and rectification) are present. In comparison, models in which center and surround share identical kinetics perform worse, even when RRP nonlinearity is retained (DSI $\leq 10\%$).

To eliminate ambiguity about the linear interactions, I extended the Borst model to include explicit spatial RFs and surrounds. The full implementation (<200 lines of code; roughly half is plotting) is available at:

<https://github.com/PolegPolskyLab/DS-mechanisms/tree/Dec2025>

minimal_example.py

Regarding Figure S13 (now S14): this figure is intentionally schematic. Its purpose is to illustrate the conceptual distinction between the sensor (photodetector/RF) and the connection to the detector (wire/synaptic release). It is not meant to depict biophysical detail but rather to highlight the key point that, although the signals arriving at the detector must differ in their temporal envelopes to generate DS, the specific implementation at the sensor level can vary widely.

The figure serves as a simplified, didactic contrast (an artificial “instrument analogy”) to communicate this high-level concept. Importantly, the revised manuscript now includes a much more rigorous and explicit mechanistic analysis of the linear and nonlinear components that generate DS (Figs. S5–S6). These new analyses directly address the reviewer’s concerns by demonstrating precisely how different RF and synaptic components contribute to temporal asymmetries.

Given this expanded, detailed treatment of the mechanisms, I believe Fig. S14 continues to serve its intended clarifying role and can remain unchanged.

Conclusion

Both linear and nonlinear architectures can generate DS for moving bars and drifting sinusoidal gratings.

- Kinetic models shift the response phase directly and therefore do not require nonlinear elements, although, as shown in Figs. S5-S6, nonlinearities substantially improve DSI.
- Models constrained to vary only RF amplitude or spatial extent cannot directly manipulate phase, but can nonetheless generate DS through linear center-surround interactions when the two components have distinct temporal properties.
- Biologically grounded nonlinearities (RRP dynamics, center-surround rectification) significantly enhance DS but are not strictly required for the computation.

Reviewer #2 (Remarks to the Author):

My major points 1, 2, and 4 were sufficiently addressed.

My point 3, about biological feasibility was dismissed with no changes to the manuscript that I can discern. I don't think I'm being unreasonable, and another reviewer also raised a concern about the claims of biological feasibility. To be clear, it isn't that I believe that

diverse spatial and temporal tuning properties in upstream neurons is non-biological. But rather than in the present manuscript the DS that is observed in some models appears to rely on very specific, non-monotonic, or otherwise "strange" arrangement of those inputs. The author's response suggests that the particular arrangement doesn't matter, and that there are families of solutions that have different arrangements that work perfectly well. If this is the reason for the apparent non-biological flavor of some of the models presented in the main figures then I believe this should be at least demonstrated and explained to the reader, given the paper's strong claims about biological feasibility.

I agree with the reviewer's point. Accordingly, I modified most of the example implementations so that the simulations now produce monotonic functions. Achieving monotonicity is straightforward in models where only one or a few parameters are allowed to vary. However, when the parameter space becomes large, models frequently converge on non-monotonic functions. This behavior is expected, for the reasons outlined below.

All RGC models in the manuscript are innervated by four distinct groups of excitatory cells and, in some cases, an additional four groups of inhibitory presynaptic cells. When no explicit constraints are imposed, each presynaptic group can evolve independently. In such a high-dimensional parameter space, a given algorithmic solution can be realized in multiple, equally valid structural implementations. For example, in several models the optimal solution was effectively to silence a subset of inputs. The models can accomplish this in several interchangeable ways, including: (1) generating strong, prolonged surround suppression that shunts center activation; (2) evolving center and surround RFs with matching kinetics and spatial parameters, thereby canceling each other; or (3) reducing the synaptic weights from those presynaptic cells to zero.

Because these presynaptic populations operate independently, any of these solutions may arise randomly yet produce the same functional outcome. Furthermore, when synaptic weights for a presynaptic group are driven to zero, the remaining RF parameters of that group become arbitrary, as they no longer influence the postsynaptic cell's response. Consequently, in sufficiently complex models, examining a single parameter dimension (e.g., RF kinetics alone) may not yield a monotonic dependence. Instead, the appropriate approach is to evaluate the model's complete functional output, for instance, the postsynaptic cell's final response. When viewed at this level, the underlying computational logic becomes clear and follows the computational primitives described in the main text.

Overall, among the ~60 circuit implementations presented in the manuscript, only about five exhibit non-monotonic parameter dependencies, all of which arise from the complex high-dimensional scenarios described above.

Reviewer #2 (Remarks on code availability):

I could not get main.py to run. I suggest making a requirements.txt file for installation, or making the repository an installable python package, and adding a readme with installation and basic entry point instructions.

Both requirements.txt and readme.txt should be present. NEURON can be finicky to install. I suggest using Google Colab, where the installation of NEURON always works.

```
#install neuron for python
!pip install neuron -q
# clone the github
!git clone --branch Sep2025 --recurse-
submodules https://github.com/PolegPolskyLab/DS-mechanisms.git
#install requirements
%pip install -U pip
%pip install -r requirements.txt

# compile the neuron mod files
%cd DS-mechanisms/mod
!nrnivmodl

# copy the x84_64 folder to the main directory
!mv -v x86_64 ..
%cd ..

# this should now work!
!python main.py
```

In addition, I created a simplified version of the model based on (Borst, 2018), showing how DS can be created in different RF configurations in response to drifting gratings stimulation. It can be accessed from:

<https://github.com/PolegPolskyLab/DS-mechanisms/tree/Dec2025>
minimal_example.py

Reviewer #3 (Remarks to the Author):

The manuscript has improved. Especially useful was the addition of comparison between moving bars and gratings. However, in the second reading of the manuscript, the work does strike as delivering new conceptual advance. My take-away message from the manuscript is that directional selectivity can be achieved in any system that possesses asymmetry in its receptive field shape and/or dynamics. If that is correct, perhaps it is useful to state this up front. I also agree with other reviewers' concerns, especially those from Reviewer 4.

This is accurate and I modified the text to reflect this insight better

Reviewer #4 (Remarks to the Author):

I appreciate the authors' careful and detailed responses to the initial review. Below I offer a few additional comments on several key issues.

First, my earlier concern was not about the breadth of the topic per se. It is important not to conflate a "specific" research goal with a "narrow" one. A study can address a large, foundational question and still define a clear, specific objective; conversely, even a very narrow topic can be presented in a vague or diffuse manner.

Second, it remains unclear how the reported results can be meaningfully linked to actual brain mechanisms. What concrete experimental approaches could be used to explore such links, and what evidence supports the claim that the reported results are biologically plausible? Stating that the model is "constructed from known circuit components" does not by itself guarantee relevance to real neural systems. In a design that is not centered on testing a sharply defined hypothesis but instead on simulating a moderately realistic network and searching for interesting outcomes, it is difficult to know how specific modeling choices or constraints influence the results. Moreover, the model conditions used here — shallow architecture, random initialization, etc. — differs substantially from biological circuits, which makes strong claims about biological reality hard to accept.

Relatedly, the manuscript asserts that the work "reveals four new computational primitives distinct from the canonical H&R and B&L models." Such predictions would indeed be exciting if the model could further lead to experimental observations in the brain. However, producing novel patterns from a partially biologically inspired network is always possible, and without stronger justification it is not clear that other investigators should be motivated to invest in experimental validation.

The authors summarize their approach as "both hypothesis-driven and predictive," with the stated hypothesis that "DS is supported by implementations and algorithmic

strategies beyond the canonical models.” As written, this hypothesis is so general that it is effectively unfalsifiable; it would be surprising if any reasonable dataset failed to support it. This makes the study appear less truly hypothesis-driven and more exploratory or post hoc in nature.

The outcome — rediscovery of classical models and identification of new computational primitives — does not convincingly advance the field, as validation of classical models has been addressed extensively in prior work.

What advances the field (among others) is the identification of new computational primitives.

Likewise, the prediction of “experimentally testable mechanisms” leaves open the key question of whether the newly proposed primitives are genuinely expected to exist and be observable in the brain, and if not, why not.

In summary, while the modeling framework is interesting, the manuscript would benefit greatly from a clearer articulation of specific, falsifiable hypotheses, stronger justification for the biological plausibility of the proposed mechanisms, and a more rigorous discussion of how the predictions could be tested experimentally.

DS models:

It is possible to make a model of the DS system that is complex enough, which would have multiple solutions that are not particularly interesting because they lack biological relevance. For example, deep neural networks can reach multiple solutions because of the huge parameter space one can cover, but the majority of these solutions will have nothing to do with biology and instead reflect abstract mathematical properties of the network.

Deep neural networks can approximate any function, this generality is precisely why they are not well-suited for the present question. Demonstrating that a sufficiently deep network can compute DS would add little insight: such networks can implement virtually any mapping. Instead, the models I propose here are concise and highly constrained throughout the main text of the manuscript. I do explore “unconstrained” shallow models – but these are only mentioned briefly and presented in supplementary materials for a reason: there is not that much information one can glean from these complicated solutions.

Confusingly, going to deep networks is exactly what the reviewer is suggesting. If the reviewer means to explore a fully biologically realistic deep model that represents processing in the retina, LGN, V1, and beyond, this is an enormously complex undertaking, well beyond the scope of the current work and in practice only feasible with resources on the scale of the Blue Brain project. More importantly, as I showed in the

first rebuttal, even in a simplified two-layer system, the solution space is already extremely rich; adding additional layers would expand this space exponentially and obscure, rather than clarify, the underlying computational principles.

I chose a shallow model architecture for two main reasons. First, it aligns with the classical frameworks of Hassenstein & Reichardt and Barlow & Levick, which operate at the same level of abstraction. Contrary to the reviewer's suggestion, a shallow architecture is an appropriate approximation for the retinal circuit composed of bipolar cells synapsing onto direction-selective ganglion cells, and it captures the essential computational transformations known to occur at this stage. The revised main text now includes this information.

Second, in contrast to the reviewer's assertion, the models I am presenting are experimentally tractable. My organization of the manuscript follows logically from the RF composition and lends itself to simple experimental validation that requires only measurement of the RF structure. Another major advantage of simplifying, rather than complicating the analysis, is the concrete, testable model predictions (see more about these critical points below).

The reviewer complains about randomized parameter initialization. I think they are missing a key point. All the parameters are still constrained within the biologically relevant range – it is only where within the range they are instantiated that the randomization is applied (this is now explicitly mentioned in the revised main text). Naturally, I spend much effort to describe the solutions reached in the simulations. That doesn't mean that such solutions are common. The opposite is true. DS in the first generation, composed of randomly initialized models, is invariantly low. Effective solutions occupy a tiny fraction of the parameter space; this is true for classical models and for the new computational primitives and circuit implementations described here.

Experimental evidence:

The central conceptual finding of this project is that basic RF components beyond kinetics can robustly support DS computation. The manuscript provides specific, testable predictions. For example, the models predict that DS should be enhanced when presynaptic RF sizes increase progressively along the sequence of activation in the preferred direction. This prediction emerged consistently across all models constrained to vary only in RF center size, even when initialized with random values - strong evidence that this configuration represents a uniquely optimal solution under those constraints. Random initialization was essential for discovering this convergence.

To address the reviewer's concern about biological plausibility, I now cite (Matsumoto et al., 2019) in the revised Discussion. In that study, the authors examined the temporal dynamics of bipolar cell inputs to direction-selective ganglion cells and reported clear

differences in temporal kinetics consistent with a Hassenstein-Reichardt-type computation. As part of their basic characterization of visual response properties, they also measured the spatial extent of the receptive fields. **They briefly mention that RF size increases along the preferred direction of activation.** This observation was not pursued further in their work, as the prevailing view in the field holds that RF size does not contribute to DS tuning.

For clarity, I reproduce their Fig. 2F below. As shown, there is both a sharpening of the temporal RF and a widening of the spatial RF in the preferred direction (upward in their convention, matching the convention used in my work). Note that the yellow signal most likely originates from glutamatergic amacrine cells, and mediates a small fraction of the excitatory drive to the cell.

I compare these findings directly with predictions from my models. In panel (a), the left and middle plots correspond to models in which only kinetic parameters or only spatial RF size can vary (as shown in Fig. 1). The right plot shows a model in which both parameters can vary simultaneously. In panel (b), I present the resulting DSI values across stimulus velocities. As predicted, the combined model performs better than

models restricted to varying only kinetics or only RF size.

The alignment is striking: the experimental results in (Matsumoto et al., 2019) mirror the qualitative and quantitative predictions of my models.

With this information and given the long history of receptive-field mapping in visual neuroscience, I am unsure why the reviewer finds the proposed “concrete experimental approaches” unclear. Techniques to measure RF size, structure, and dynamics are standard and widely used, and I outline how well-established RF-mapping approaches, used in vision science for over 70 years, can be employed to test these simple predictions experimentally (Fig. 9).

Testable predictions and hypotheses:

The familiar maxim “all models are wrong, but some are useful” is particularly relevant here. Every theoretical model necessarily simplifies biological reality. In this work, I present models at two different levels of abstraction and across two different systems. As detailed above, I have addressed the reviewer’s concerns regarding experimental validation. However, contrary to the reviewer’s suggestion, I believe the simplified models are often more valuable than attempts to fully replicate biological complexity, an undertaking that may not even be achievable in practice. Simplified models allow for clarity, systematic dissection, and direct insight into the underlying computational principles.

For instance, I predict that an anti-Barlow & Levick (anti-B&L) algorithm exists in the brain. The computation itself is conceptually straightforward: inhibition that precedes excitation in the preferred direction. Observing this computation in highly simplified models is not a weakness; rather, it demonstrates the generality and robustness of the underlying algorithm.

This perspective is consistent with the history of the canonical B&L model. When first described in the mammalian retina (Barlow & Levick, 1965), the model operated at a comparably abstract level. Only later did experimental work reveal that starburst

amacrine cells exhibit the spatially offset RFs and slower dynamics predicted by the model. Importantly, the same conceptual algorithm applies across diverse systems, including mammalian cortex and the *Drosophila* visual system, despite the absence of starburst cells in those circuits. The implementation differs, but the computation is conserved (see instance (Borst & Helmstaedter, 2015)). Thus, models that are not tied to detailed biological realism can still be highly informative and broadly applicable.

My goal in this work is to discover novel algorithms and implementations; naturally, there is not yet direct experimental evidence for all of the models presented. That does not diminish their scientific value. To suggest that predictions should only be studied once experimental evidence already exists would be to eliminate the role of theory in advancing neuroscience. The reviewer writes that “such predictions would indeed be exciting if the model could further lead to experimental observations in the brain.” But this is precisely the point: the model generates experimentally testable predictions.

Below, I list several of these concrete, falsifiable predictions along with the experimental approaches (using established and widely available methods) that could be used to test them:

Prediction	Example experimental approach
To maximize DS, RF size should increase in the preferred direction	Model validated experimentally by recording glutamatergic signals to direction selective ganglion cell.
Synaptic weights by themselves can promote DS in two distinct configurations.	Model validated experimentally by postsynaptic patch clamp recordings
Differences in RF surround size, kinetics, and amplitude can promote DS in two different implementations each, and these implementations require different combinations of RF properties	Surround measurements can be performed directly from presynaptic cells or using analysis of glutamate signals onto the postsynaptic cell.
A shallow network with excitatory inputs only can combine the responses of the presynaptic inputs in four distinct ways, or in other words, four algorithms exist for excitation-only DS (one was known previously).	Calcium or glutamate imaging to record the spatiotemporal patterns of the excitatory input.
In a system with an inhibitory drive, the amplitude, kinetics and size of the center and surround RF components of the inhibitory drive can be combined in specific ways (which I explore at length in the manuscript) to generate DS	Surround measurements can be performed directly from presynaptic cells (using calcium imaging) or by analyzing GABAergic signals onto the postsynaptic cell.
At least two new algorithmic solutions to how excitation and inhibition can interact to generate DS.	These algorithms can be tested by comparing the temporal dynamics of the excitatory and inhibitory drives in

	preferred and null directions using postsynaptic voltage clamp or calcium imaging of presynaptic (E and I) populations.
--	---

Further, I respectfully disagree with the assertion that an explicitly framed falsifiable hypothesis is essential to appreciate the contribution of this work. Many influential theoretical studies in neuroscience were not organized around a single, narrowly formulated hypothesis, yet they clearly generated testable (and ultimately falsifiable) predictions.

A classical example is Rall's work on dendritic integration (e.g., (Rall, 1967)), in which mathematical models were used to analyze signaling dynamics in cells with dendrites. Rall derived numerous predictions about attenuation and filtering in passive dendrites, many of which were later confirmed experimentally. It is not obvious what a concise "falsifiable hypothesis" would have been for this entire body of work, beyond something trivial along the lines of "dendrites influence somatic responses," which adds little conceptual value and does not capture the depth of the theoretical advances.

Similarly, in the next generation of computational studies inspired by Rall, Mainen & Sejnowski (Mainen & Sejnowski, 1996) applied cable theory to models with active conductances and showed that differences in dendritic morphology can produce markedly distinct firing patterns, even when the underlying ion channels are similar. The key insight is that morphology and compartmental integration shape nonlinear interactions with voltage-gated channels. Here again, while their simulations produced many testable predictions, the overarching "hypothesis" might be summarized as "morphology affects function," which, as the reviewer notes in another context, is so broad that it is essentially unfalsifiable.

In all of these examples, the central scientific question is not "whether" but "how." Long before Rall, neuroscientists knew that dendrites affect signal spread. Mainen & Sejnowski were not recognized because they were the first to propose that dendritic structure can influence action potential generation, but because they elucidated specific mechanisms by which it does so.

In the same spirit, my manuscript focuses on how direction selectivity can arise from particular receptive-field structures and circuit motifs. I provide concrete examples of how newly identified algorithms can be implemented in biological circuits, offer higher-level conceptual frameworks for understanding them, and specify the detailed requirements for these computations. All of this is expressed in precise mathematical and computational terms that yield clear, testable predictions, some of which already

align with existing data. In this sense, the work follows the tradition of theoretical studies that advance mechanistic understanding even when a single, neatly packaged falsifiable hypothesis is not the primary organizing principle.

More broadly, theoretical work plays a crucial role in identifying candidate mechanisms that may otherwise remain unexplored. Publishing these predictions provides the field with concrete hypotheses, quantitative expectations, and clear experimental readouts that can motivate and guide empirical studies. Without such a theoretical framework, many of these potential mechanisms would simply not be considered or tested. In this sense, the work is intended to stimulate and support future experimental efforts, not replace them.

In summary, the manuscript provides multiple testable predictions, at both the implementation and algorithmic levels, regarding DS computation. Available experimental data already support several of these predictions. Furthermore, I outline how well-established, widely used RF-mapping methods can be used to test the additional predictions generated by this work.

References:

- Barlow, H. B., & Levick, W. R. (1965). The mechanism of directionally selective units in rabbit's retina. *J Physiol*, 178(3), 477-504.
http://www.ncbi.nlm.nih.gov/entrez/query.fcgi?cmd=Retrieve&db=PubMed&dopt=Citation&list_uids=5827909
- Borst, A. (2018). A biophysical mechanism for preferred direction enhancement in fly motion vision. *PLoS Comput Biol*, 14(6), e1006240.
<https://doi.org/10.1371/journal.pcbi.1006240>
- Borst, A., & Helmstaedter, M. (2015). Common circuit design in fly and mammalian motion vision. *Nat Neurosci*, 18(8), 1067-1076. <https://doi.org/10.1038/nn.4050>
- Mainen, Z. F., & Sejnowski, T. J. (1996). Influence of dendritic structure on firing pattern in model neocortical neurons. *Nature*, 382(6589), 363-366.
<https://doi.org/10.1038/382363a0>
- Matsumoto, A., Briggman, K. L., & Yonehara, K. (2019). Spatiotemporally Asymmetric Excitation Supports Mammalian Retinal Motion Sensitivity. *Curr Biol*.
<https://doi.org/10.1016/j.cub.2019.08.048>
- Rall, W. (1967). Distinguishing theoretical synaptic potentials computed for different soma-dendritic distributions of synaptic input. *J Neurophysiol*, 30(5), 1138-1168.
<http://www.ncbi.nlm.nih.gov/pubmed/6055351>
- <https://www.physiology.org/doi/pdf/10.1152/jn.1967.30.5.1138>

I thank all the reviewers for their time and effort. I believe the critiques greatly improved the work.

Reviewer #1 (Remarks to the Author):

The author has done a fine and careful job addressing my comments. I especially appreciated Figs S4 and S5, which lay out clear where direction selectivity arises for sinusoids. I have only minor comments.

Minor comments

1) Overall, I wonder if there are a few places where this paper might benefit by adopting the terminology of separable/inseparable linear filtering and oriented space-time filter – the terminology from Adelson and Bergen that has been so helpful in that part of the field. I'm thinking especially of places like around line 186, where my reading is that you are saying (for the sinusoids) that you can create oriented space-time RFs by combining multiple inseparable space-time filters. (Though it's more complicated because of the ReLU on each c-s synaptic input, so the simple linear analysis of the motion energy model doesn't necessarily hold.) For the simplified linear models, which are equivalent to the ME linear filtering step, this terminology might be especially apt.

I agree, the revised text includes this language when I describe Figure 8.

2) Line 150: "anti-HR" is being defined as the neurons with longer responses on the null-direction side of the RF. But kinetics are identical, so the longer responses are stimulus dependent. This only looks "anti-HR" with bars as the stimulus. With sinusoids, it would not. Shouldn't the listed properties be properties of the circuit, not the stimulus-circuit pairing?

Anti-HR is possible with sinusoids. The anti-HR properties would be evident as changes in the direction of the phase of the response. However, due to the cyclic nature of the stimulus, the effect is somewhat complicated to analyze, as phase-induced changes with HR mechanism may mimic the anti-HR mechanism.

3) Last, I remain a tad incredulous at the claim near line 404 that these numerical models are "simpler" than classical frameworks like the HRC and ME. The reasoning given is that the numerical models are simpler because each model configuration required independent testing. I cannot see how one follows from the other. I think there are many things you could say about the differences between these numerical models and the kind of analytically tractable models represented by the HRC and ME. The HRC

and ME models are not biologically realistic, they are in fact overly simplified, so that the nonlinearities and phases and anti-symmetric subtractions are prescribed, rather than allowed to be fit parameters. For instance, the original HRC model, proposed in the 1956 paper, has two parameters: the distance between sensors and the time delay. It has a very restricted architecture to make up for the lack of parameters and to focus explicitly on the pairwise correlations that are the simplest hallmark of motion. However, I don't think one can argue that it's not simple. Overall, I think it would be worthwhile to spell out precise differences and how they contributed to the new findings, rather than arguing that these numerical models (with nonlinear synaptic release dynamics and integrated neural morphology!) are somehow just 'simpler', which I think is hard to define, hard to justify, and an unnecessarily controversial claim to make.

I agree, the revised text labels the models as 'comparable'.

Minor minor comments

- 1) Line 17 in abstract: "previously undescribed" instead of "newly discovered"?
- 2) Line 37: "computational motifs" rather than principles?
- 3) Line 79: "nonlinear RRP dynamics"? It might be worthwhile to emphasize linear and nonlinear operations throughout...
- 4) Line 115 and beyond: RF size gradient with identical positive only kinetics gets bar DS, but NOT sinusoidal DS. Perhaps some telegraphing of this later caveat is order?
- 5) Line 315: for HR+BL formulation, probably want to also cite Leong ... Clandinin 2016 JNeuro.
- 6) Line 386: caveat also: DS is for specific stimuli; paper now gives examples for detectors that work well for bars but not at all for sinusoids. This seems like a point to highlight. What stimuli should we all be using to measure DS? More of them?
- 7) Line 378: I didn't see a 'Second, ...' for enumerating clearly the second key insight.
- 8) Line 452: Author suggests that sinusoids are more complex than simpler bars. However, these are just two representations of space (regular and Fourier) and I'm not sure there's a good reason to claim that one is simpler or more complex than the other. If anything, most of the basic motion detection behaviors seem to be for detecting flow fields and stabilizing eyes and bodies – extended stimuli seem better suited for investigating/optimizing such detectors.
- 9) Line 456: Gruntman did not observe DS inhibition, since that does not appear to be a component of the fly DS circuit, so shouldn't be cited in this list.
- 10) Supp 14: In b, one could mention that this works as drawn for some stimuli (like bars) but not others, like sinusoids, unless you have spatiotemporally inseparable filters for p1 and p2? Consistent with Supp Fig 5.

I made the changes to address all the minor concerns raised by the reviewer

Reviewer #2 (Remarks to the Author):

All of my concerns have been addressed. Thanks for the thoughtful responses.

Reviewer #3 (Remarks to the Author):

The author have adequately addressed the previous comments. The revised discussion clearly describes how the computational predictions made here can be tested in future studies, and how the existing experimental evidence on receptive field asymmetries provides early support for the predictions made here.

Reviewer #3 (Remarks on code availability):

Well, I have tried to install the code but it did not run on my machine. It is most likely a problem with my python environment. So, I cannot state that the code is not valid.

Reviewer #4 (Remarks to the Author):

The efforts made by the author to address the concerns raised by the reviewers are appreciated. While the revised manuscript and rebuttal may give the impression of engaging with these issues, many of the added explanations appear to stem from a misunderstanding of the underlying questions and are therefore largely irrelevant to the concerns raised.

As emphasized by other reviewers, the core issue remains the need to clearly establish the biological relevance of the findings — specifically, how the model's predictions relate to actual biological principles. While the model may produce simulation results that align with experimental data, this does not necessarily imply that it accurately reflects biological mechanisms, nor does it automatically confer scientific significance. Such significance typically arises from the formulation and validation of clearly falsifiable hypotheses, and this aspect appears to remain underdeveloped in the current manuscript.